# Generalized Top-$k$ Mallows Model for Ranked Choices

**Shahrzad Haddadan**
Rutgers Business School
Piscataway, NJ
shaddadan@business.rutgers.edu

**Sara Ahmadian**
Google research
Seattle, WA
sahmadian@gmail.com

## Abstract

The classic Mallows model is a foundational tool for modeling user preferences. However, it has limitations in capturing real-world scenarios, where users often focus only on a limited set of preferred items and are indifferent to the rest. To address this, extensions such as the top-$k$ Mallows model have been proposed, aligning better with practical applications. In this paper, we address several challenges related to the generalized top-$k$ Mallows model, with a focus on analyzing buyer choices. Our key contributions are: (1) a novel sampling scheme tailored to generalized top-$k$ Mallows models, (2) an efficient algorithm for computing choice probabilities under this model, and (3) an active learning algorithm for estimating the model parameters from observed choice data. These contributions provide new tools for analysis and prediction in critical decision-making scenarios. We present a rigorous mathematical analysis for the performance of our algorithms. Furthermore, through extensive experiments on synthetic data and real-world data, we demonstrate the scalability and accuracy of our proposed methods, and we compare the predictive power of Mallows model for top-$k$ lists compared to the simpler Multinomial Logit model.

## 1 Introduction

User preferences over a set of alternative items play a crucial role in various decision-making scenarios. A key concept in this context is the *choices* customers make when presented with a subset of alternatives, referred to as an *assortment*, drawn from a larger pool of items. The mathematical modeling of preferences and choices is both essential and challenging. It enables researchers and business leaders to analyze and predict customer behavior, thereby informing more effective decision-making. Probabilistic models over rankings, such as the Plackett-Luce (PL) and Mallows models, are widely used to represent user preferences. Based on the Plackett-Luce (PL) model, the Multinomial Logit (MNL) model has been suggested for modeling choice, and it has been extensively applied in choice modeling due to its simplicity and interpretability.

The Mallows model (Mallows, 1957) is a distance-based probability distribution defined over permutations and is reminiscent of the Gaussian distribution for scalar variables. It has been used successfully to model preferences, particularly where ranking data is available. Recent work has demonstrated the high predictive power of the Mallows model in modeling choices (Désir et al., 2023, 2021), sparking follow-up research into revenue management and assortment optimization in this framework Désir et al. (2021, 2023); Feng & Tang (2022); Rieger & Segev (2023). One key challenge in applying the Mallows model more broadly is that its classic definition is restricted to full permutations.

In many real-world scenarios, however, user preferences are often observed as *top-$k$ lists* rather than full rankings. For example, many platforms display only a subset of items to customers and ask them to rank a fixed number of preferred items. In recommender systems, advertising platforms, search engines, news aggregators, and social media friend suggestions, users are typically shown only the

top-$k$ most relevant items, rather than an exhaustive list. Likewise, users often express preferences for a limited number of favorite items and show indifference toward the remainder.

This recurring structure in many applications has motivated the extension of the Mallows model to handle top-$k$ lists. The design of algorithm for this variant is significantly more complex than in the traditional permutation-based setting, leading to a growing body of research focused on learning, inference, and aggregation under the top-$k$ Mallows model (Lu & Boutilier, 2011; Chierichetti et al., 2018a; Collas & Irurozki, 2021; Vitelli et al., 2018; Fotakis et al., 2021; Boehmer et al., 2023; Goibert et al., 2023; Awadelkarim & Ugander, 2024; Qian & Philip, 2019; Akbari & Escobedo, 2022).

In this paper, we propose employing a generalized Mallows model for top-$k$ lists for choice modeling, and we develop efficient algorithms related to generating samples from it, learning its parameters and finding choice probabilities. Our approach addresses more realistic preference structures, where users are unlikely to hold complete rankings over a large set of items, an assumption that is often impractical in real-world applications.

## 1.1 Related Work

In this section we present relevant related work on choice modeling and Mallows model.

**Choice modeling**  Various probabilistic models have been developed to capture choice behavior, with the Multinomial Logit (MNL) (Bradley & Terry, 1952) model being the most widely used due to its simplicity and interpretability. In the MNL model each product is assigned a positive score or weight, and the probability of selecting a product from an assortment is proportional to its score. Importantly, the MNL model satisfies Luce's Choice Axiom, also known as the Independence of Irrelevant Alternatives (IIA). While this property makes the MNL model analytically convenient, it also limits its expressiveness in capturing more complex choice behaviors.

To overcome the limitations of MNL, the mixture MNL model (also known as mixed logit) was introduced and popularized by McFadden & Train (2000). Learning the parameters of a mixed MNL model from observed choices—where each observation is an assortment and a chosen item—is a key challenge. Early approaches used heuristics based on maximum likelihood estimation (Dempster et al., 1977), and more recent work has provided statistically rigorous methods with provable sample complexity guarantees (Chierichetti et al., 2018b; Oh & Shah, 2014; Ma et al., 2022).

Beyond MNL-based models, other frameworks such as the Mallows-based choice model (Désir et al., 2021) and Markov chain-based models (Blanchet et al., 2016) have been proposed. These models offer greater flexibility but come with added complexity: tasks that are straightforward in MNL, such as computing choice probabilities or generating samples become substantially more challenging.

**Mallows model**  The Mallows model (MM), originally introduced by Mallows (1957), defines a distance-based distribution over full rankings, where the probability of a permutation decays exponentially with its distance from a central (ground-truth) ranking. This property has made MM a useful foundation for preference modeling in machine learning. To better accommodate real-world data—where users often provide partial rather than full rankings—several extensions have been proposed. Early work by Fligner & Verducci (1986) and Lebanon & Mao (2008) adapted MM to top-$k$ rankings. Later Chierichetti et al. (2018a) proposed a new distance measure on top-$k$ lists and defined a parametrized Mallows model for top-$k$ lists based on it (TopKMM).

One of the strengths of the standard Mallows model on permutations is the closed-form expression of its normalization constant which enables the design of several algorithms. For instance, the Repeated Insertion Method (RIM) (Doignon et al., 2004) allows efficient sampling from the full-ranking Mallows model. However, this tractability does not extend to TopKMM: computing probabilities or generating samples becomes non-trivial due to the lack of a closed-form normalizing constant. While Chierichetti et al. (2018a) proposed a dynamic programming approach to generate samples, they highlighted the absence of an RIM-like sampling method for top-$k$ lists as an open problem.

Désir et al. (2021) proposed leveraging the Mallows model to capture choice behavior, demonstrating improved predictive accuracy over traditional models like the Multinomial Logit (MNL). However, a key challenge in applying the Mallows model to choice modeling lies in computing choice probabilities—i.e., the likelihood that a given item is selected from an assortment. Unlike MNL, where such probabilities have closed-form solutions the Mallows model does not admit such tractable computation in general. To address this, some works have introduced Mallows-like models designed to simplify choice probabilities calculations (Feng & Tang, 2022). In contrast, Désir et al. (2023)

tackle the original model directly and develop a dynamic programming approach for computing choice probabilities by leveraging key ideas form the Repeated Insertion Method (RIM).

Learning the parameters of the Mallows model or its mixtures from full permutation, or top-$k$ list[1], samples has been extensively studied in prior work (Liu & Moitra, 2018a; Braverman & Mossel, 2008; Awasthi et al., 2014; Seshadri et al., 2020; Tang, 2019; Collas & Irurozki, 2021; Akbari & Escobedo, 2022; Liu & Moitra, 2018b; Chierichetti et al., 2015). Some studies focus on learning these parameters from historical data, while others consider an *active learning* setting, where upon each consumer's arrival, the platform adaptively selects the assortment of items to offer based on past observations Susan et al. (2022). In contrast, learning model parameters from partial observations presents a greater challenge. Several works address this problem by studying learning from pairwise comparisons Lu & Boutilier (2014); Vitelli et al. (2018); Tang (2019). A less explored problem is estimating the model parameters when only the users' choices from offered assortments of arbitrary sizes are observed. Existing methods in this setting often lack finite-sample complexity guarantees, limiting their theoretical robustness.

## 1.2 Summary of Contributions

In this paper, we focus on Mallows model on top-$k$ lists, and we make several algorithmic contributions for the usage of this model. In order to obtain our results in full generality, we extend Chierichetti et al. (2018a)'s model on Mallows model for top-$k$ list to a generalized version by associating a weight to each product. Our algorithmic contribution are listed as follows:

1. *Sampling:* PROFILE BASED REPEATED INSERTION METHOD (PRIM): The *Repeated Insertion Method* (RIM) is a common method for sampling permutations in the classic Mallows model. However, extending this method to the top-$k$ Mallows model remains an open problem. Our proposed algorithm, PRIM, offers similar functionality to RIM and improves upon the prior dynamic programming approach by Chierichetti et al. (2018a), reducing the runtime from $O(k^2 4^k + k^2 \log n)$ to $O(k2^k + k^2 \log n)$ .

2. *Choice Probabilities:* DYNAMIC PROGRAMMING FOR CHOICE PROBABILITIES (DYPCHIP): DYPCHIP is an algorithm to calculate the choice probabilities when they are inferred from a Mallows model on top-$k$ lists. This result extends the work of Désir et al. (2021) who consider the classic Mallows model for permutations.

3. *Learning the Center:* BUILD CENTER FROM CHOICES (BUCCHOI): BUCCHOI is an *active learning algorithm* designed to learn the center of a top-$k$ Mallows model distribution. It operates by presenting assortments of a specified size $r$ to customers and, based on their observed choices, infers both the ranking of the center and the size of the center $k$.

The accuracy and complexity of these algorithms are demonstrated through rigorous mathematical analysis as well as experiments on real-world and synthetic data.

Furthermore, we apply these algorithms and fit a top-$k$ Mallows model to a real-world publicly available dataset including users' preferences over 100 sushi types, represented as top-10 lists (Kamishima et al., 2005). This model helps us predict choice probabilities with high accuracy, and our results demonstrate that the top-$k$ Mallows model achieves significantly higher predictive accuracy than the Multinomial Logit model on this dataset

## 2 Preliminaries and Definitions

Let $N = [n] := \{1, 2, \cdots, n\}$ represent a universe of $n$ elements. A top-$k$ list is a partial order on $N$ structured as $i_1 \succ i_2 \succ \cdots \succ i_k \succ \{i_{k+1}, \cdots, i_n\}$ , where the top-$k$ elements are strictly ordered, while the remaining $n - k$ elements are incomparable to each other. The collection of all top-$k$ lists over $N$ is represented by $T_{k,N}$, where $T_{n,N} = S_N$ corresponds to the symmetric group on $N$.

For a top-$k$ list $\tau$ and a position $l \in [k]$, $\tau(l)$ refers to the element ranked at position $l$, while $\bar{\tau}$ denotes the set of elements ranked below the top-$k$. For simplicity, we sometimes use $\tau$ to represent the top-$k$ elements $\{\tau(1), \tau(2), \ldots \tau(k)\}$. Thus, $i \in \tau$ indicates that $i$ is ranked among the top-$k$ elements of $\tau$, and set operations like $\subseteq$ and $\cap$ are applied accordingly. For $i, j \in N$, $i \succ_\tau j$ means $i$ is ranked

---

[1]We remark that some prior work considers scenarios where users select multiple items or a list of $k$ items—referred to as a top-$k$ choice. In contrast, our focus is solely on the selection of a single item, and we use the term *top-$k$* only as a parameter of the Mallows model.

above $j$ in $\tau$, i.e., $i \in \tau$ and either $j \in \bar{\tau}$ or $j \in \tau$ but ranked below $i$. Additionally, $i \parallel_\tau j$ indicates $i$ and $j$ are incomparable (both are in $\bar{\tau}$), while $i \perp_\tau j$ means they are comparable ($i \succ_\tau j$ or $j \succ_\tau i$).

In this paper, we utilize the widely recognized Kendall's Tau distance, a commonly used metric that quantifies the number of pairwise disagreements between two permutations (Fagin et al., 2003; Critchlow, 2012). This concept has been extended to top-$k$ lists, where it no longer forms a true metric but retains useful mathematical properties. Given a parameter $p \geq 0$, the $p$-parametrized distance between $\tau, \tau' \in T_{k,N}$ is defined as

$$\mathcal{K}^p(\tau, \tau') = \sum_{i,j \in \tau \cup \tau' : i < j} \mathcal{K}^p_{i,j}(\tau, \tau'), \text{ where } \mathcal{K}^p_{i,j}(\tau, \tau') = \begin{cases} 1 & \text{if } (i \succ_{\tau'} j \ \& \ j \succ_\tau i) \text{ or vice-versa} \\ p & \text{if } (i \perp_{\tau'} j \ \& \ i \parallel_\tau j) \text{ or vice-versa} \\ 0 & \text{otherwise.} \end{cases}$$
(1)

Using this distance measure, Chierichetti et al. (2018a) define the *Mallows model for the top-$k$ lists*. Given a center $\tau^*$ and a decay parameter $\beta$, the probability distribution $\mathcal{D}$ over top-$k$ lists $T_{k,N}$ is defined as:

$$\mathbb{P}_\mathcal{D}\left[\tau \in T_{k,N}\right] \propto \exp\left(-\beta \, \mathcal{K}^p(\tau, \tau^*)\right) . \tag{TopKMM}$$

To simplify our notation, we assume, without loss of generality, that the center $\tau^*$ is always the identity list $1 \succ 2 \succ \cdots k \succ \{k+1, \cdots, n\}$. Therefore, we denote $\mathcal{K}^p(\tau, \tau^*)$ as $\mathcal{K}^p(\tau)$. For full rankings, where all elements are comparable, we simply use $\mathcal{K}(\tau)$.

A natural extension of this model arises when the elements have associated weights. *Generalized Mallows Model* (GMM) Fligner & Verducci (1986) considers this case for full rankings: Given a decay parameter $\beta$ and non-negative weights $w_i \in \mathbb{R}_{\geq 0}$ for each $i \in N$, the probability distribution $\mathcal{D}$ over rankings is defined as:

$$\mathbb{P}_\mathcal{D}\left[\tau \in T_{n,N}\right] \propto \exp\left(-\beta \sum_{i,j : i < j} w_i \mathcal{K}_{i,j}(\tau)\right) \tag{GMM}$$

where $\mathcal{K}_{i,j}(\tau)$ is 1 iff $\tau$ disagrees with $\tau^*$, i.e., $\tau$ ranks $j$ before $i$ for $j > i$ and 0 otherwise, as defined in (1). In this model, each disagreement contributes the weight of the item ranked higher in $\tau^*$. This formulation can be simplified by the use of inversion vectors as follows:

$$\mathbb{P}_\mathcal{D}\left[\tau \in T_{n,N}\right] \propto \exp\left(-\beta \sum_{i \in [k]} w_i I_i(\tau)\right) \text{ where } I_i(\tau) = \sum_{j : j > i} \mathcal{K}_{i,j}(\tau) = \sum_{j : j > i} \mathbb{1}\left(j \succ_\tau i\right)$$

where $\mathbb{1}$ is an indicator function that takes the value 1 when true and 0 otherwise.

The generalized mallows model has traditionally been defined only for full rankings, and we extend it to *Generalized Mallows Model for Top-$k$ lists*. In this setting, each element $j \in \tau^*$ is assigned a non-negative weight, i.e., $w_i \in \mathbb{R}_{\geq 0}$ for $i \in [k]$, along with an additional weight $w_0 \in \mathbb{R}_{\geq 0}$ for any element in $\bar{\tau}^*$. We use $\boldsymbol{w} \in \mathbb{R}_{\geq 0}^{k+1}$ to represent this collection of weights, which uses the following extension for inversion vectors:

**Definition 2.1** (Inversion Vectors of a Top-$k$ list). *Given a top-$k$ list $\tau \in T_{k,N}$, there are three components for inversion vectors: vectors $\mathbf{I}(\tau), \mathbf{P}(\tau) \in \mathbb{R}_{\geq 0}^k$ where for $i \in [k]$:*

$$I_i(\tau) = \sum_{j : j > i} \mathbb{1}\left(j \succ_\tau i\right), P_i(\tau) = \sum_{j : j > i} \mathbb{1}\left(i, j \in \bar{\tau}\right)$$

*and $Q(\tau) = \binom{k-\ell}{2}$ where $\ell = |\tau \cap \tau^*|$.*

Note that this definition is an alternative way to count the disagreement between $\tau$ and $\tau^*$ where the disagreement is always assigned to the higher-ranked element by $\tau^*$. When neither element is ranked higher ($i, j \notin \tau^*$), disagreements are assigned to $Q$. For example, for $\tau = (2, 1, 6, 5) \in T_{4,[8]}$, we have $\mathbf{I}(\tau) = [1, 0, 2, 2]$, $\mathbf{P}(\tau) = [0, 0, 1, 0]$, $Q(\tau) = 1$. $I_1(\tau) = 1$ since element 2 is ranked higher than 1 by $\tau$, $I_3(\tau) = 2$ since elements 5 and 6 are ranked higher than 3 and the same argument applies to $I_4(\tau)$. $P_3(\tau) = 1$ since elements 3 and 4 are not comparable by $\tau$ (but they are ranked by $\tau^*$) and this disagreement is assigned to element 3 as it is ranked higher than element 4 by $\tau^*$. $Q$ counts disagreements for elements not ranked by $\tau^*$, i.e., elements 5 and 6.

**Generalized Mallows Model for Top-$k$ lists**  Given the center $\tau^* \in T_{k,N}$ and parameters $\beta \geq 0$, $w \in \mathbb{R}^{k+1}_{\geq 0}$ and $p > 0$, the probability distribution $\mathcal{D}$ over $T_{k,N}$ is defined as:

$$\mathbb{P}_{\mathcal{D}}\left[\tau \in T_{k,N}\right] \propto \exp\left(-\beta\,\mathcal{K}^{p,w}(\tau, \tau^*)\right) \qquad \text{(TopKGMM)}$$

where

$$\mathcal{K}^{p,w}(\tau, \tau^*) := w_0 p Q(\tau) + \sum_{i \in [k]} w_i \cdot (I_i(\tau) + p P_i(\tau)).$$

Note that setting $w = \mathbf{1}$, we obtain Equation (TopKMM), and $k = n$, recovers Equation (GMM).

# 3 Sampling from TopKGMM

We begin by investigating the challenge of efficiently sampling from TopKGMM. Although Chierichetti et al. (2018a)'s sampling algorithms for TopKMM can easily be extended to incorporate weights as in TopKGMM, here our concentration is to develop a sampling algorithm with similar functionality to the *repeated insertion model*, which was left open (Chierichetti et al., 2018a). The theory we develop here not only helps in the design of the sampling algorithm PRIM but also plays a crucial role in the development of DYPCHIP, our proposed algorithm for choice probabilities.

**Theorem 3.1** (Sampling from TopKGMM). *For a given instance $\mathcal{D}$ of TopKGMM there exists an algorithm that efficiently samples a top-$k$ list according to $\mathcal{D}$ in time complexity $O(k \cdot 2^\gamma + k \log n)$,[2] and space complexity $O(k \cdot 2^\gamma)$; where $\gamma = \min\{k, n-k\}$.*

Similar to the RIM method for permutations (Doignon et al., 2004), the core idea of our approach involves iteratively adding elements to a partially ordered sequence until exactly $k$ elements are sampled. While this strategy results in a correct sampling scheme in the context of permutations, for top-$k$ lists it is essential to partition the sample space based on several features before iterative insertions begin. Our careful definition of inversion vectors in Equation (1) plays a crucial role here, as it allows us to focus on the behavior of elements in $[k]$ and specifically those sampled by $\tau$. This insights leads us to introduce *profiles* which represent the shared top-$k$ elements $\tau$ and $\tau^*$.

## 3.1 Profile-based TopKGMM Distribution

Let profile $S \subseteq [k]$ represent the subset of sample top-$k$ elements for a given top-$k$ list $\tau$. Formally,
**Definition 3.2** (top-k Profile). *Given a center $\tau^* \in T_{k,N}$ with corresponding TopKGMM distribution $\mathcal{D}$, we call a set $S \subseteq \tau^*$ a profile and we define $T_{\tau^*}(S)$ and its probability with respect to $\mathcal{D}$ as:*

$$T_{\tau^*}(S) = \{\tau | \tau \cap \tau^* = S\}, \quad \mathbb{P}_{\mathcal{D}}[S] = \sum_{\tau \in T_{\tau^*}(S)} \mathbb{P}_{\mathcal{D}}[\tau]. \qquad (2)$$

*When the center is clear from the context, we may simply use $T(S)$.*

For any $\tau \in T(S)$, since $Q(\tau) = \binom{k-|S|}{2}$ only depends on $S$, we simply write $Q(S)$. The inversion vector $P$ counts the number of lower-priority elements that were previously ranked strictly lower but are now incomparable. An element $j$ has a positive $P_j(\tau)$ only if it is not ranked by $\tau$, in which case its value is given by $P_j(\tau) = n - k - (k - \ell)$ as exactly $k - \ell$ elements from $\{k+1, \cdots, n\}$ are now ranked by $\tau$. Since $P$ strictly depends on $S$, we can use $P(S)$ instead of $P(\tau)$.

The inversion vector $\mathbf{I}$ is defined for elements in $[k]$, and for each element, is the number of lower-priority elements (w.r.t. the center) that are now ranked higher in $\tau$. For any element $j$ not ranked by $\tau$, i.e., $j \in [k] \setminus S$, $I_j(\tau) = |\{j+1, \cdots, k\} \cap S|$. Thus part of vector $I$, is entirely determined by $S$ and independent of $\tau$. Since $Q(S)$, $\mathbf{P}(S)$, and $\mathbf{I}_{j \in [k] \setminus S(\tau)}$ are independent of rankings among $\tau$'s elements and depend only on $S$, we compute the probability of $S$ given this information.

**Lemma 3.3.** *Given a TopKGMM distribution $\mathcal{D}$ with parameters $\beta$, $p$ and $w$, any profile $S = \{s_1, s_2, \ldots, s_\ell\} \subseteq [k]$ ($s_1 < s_2 < \cdots < s_l$) has probability $\mathbb{P}_{\mathcal{D}}[S]$ proportional to $\exp(-\beta f(S)) Z(S)$, where:*

$$f(S) = w_0 p Q(S) + \sum_{j \in [k] \setminus S} w_j (I_j(S) + p P_j(S)), \text{ and } \quad Z(S) = \binom{n-k}{k-\ell}(k-\ell)! \prod_{j=1}^{\ell} \sum_{r=0}^{k-j} e^{-\beta w_{s_j} r}.$$

---

[2]Here $O(k 2^\gamma)$ is the complexity of a pre-processing step, after that each sample can be generate at cost $O(k \log n)$.

*and it can be sampled in $O(2^\gamma k)$; where $\gamma = \min\{k, n-k\}$.*

## 3.2 Sampling Algorithm

Building on the results from the previous section and leveraging the concept of a *profile*, we now propose a method for sampling from the TopKGMM distribution (Algorithm 3). We first sample a profile $S$ with probability $\mathbb{P}_\mathcal{D}[S]$ (according to Lemma 3.3), and then we sample $\tau \in T(S)$ by inserting the elements $j \in S$ based on their contribution in inversion vectors Equation (TopKGMM)). The later step can be viewed as generalizing the Repeated Insertion Method (RIM). We refer to our method as PROFILE-BASED-REPEATED-INSERTION-METHOD (PRIM).

In PRIM, we generate a top-$k$ list proportional to its sampling probability in $\mathcal{D}$ conditioned on the fact that the common elements with the center top-$k$ ranking is $S$. Specifically, let $\ell = |S|$. We begin with an empty array $A$ and then sample $k - \ell$ elements from $[n] \setminus [k]$ uniformly at random. Then we sequentially, insert the elements in $S$ in increasing order of their priority. When processing element $s$, we insert $s$ in the current array $A$ at position $j \in \{0, 1, \ldots, |A|\}$ with probability proportional to

$$Pr(\text{inserting } s \text{ at position } j) \propto \exp\left(-\beta w_s \cdot j\right) \tag{3}$$

as $j$ is the number of inversion associated with element $s$ when it is inserted at position $j$. Note that higher priority elements do not contribute to inversion of $s$ so when their position when they are inserted later is not important for $s$. See Appendix A.1 for pseudocodes and proofs.

# 4 Choice modeling and probabilities

In this section, we focus on the problems related to *choice*. In particular we focus on the analysis of problems which help us predict the *(top) choice* of a customer from a set of alternatives, a.k.a an *assortment* using TopKGMM. After presenting definitions we design an algorithm which efficiently calculates choice probabilities having the distributions parameters. In Section 5 we study the opposite problem in this context, which is learning the center of a TopKGMM distribution by observing its choice data.

Given a set of products $[n]$, an *assortment* is any subset $\mathcal{A}$ of $[n]$. When $\mathcal{A}$ is offered to a customer, she may *choose* any of its elements or the "no purchase" option denoted by $\varnothing$. We use $N = [n] \cup \{\varnothing\}$ to denote all purchase options and correspondingly, we define $T_{k,N}$ to denote all top-$k$ lists over $N$. We assume that the preferences of customers are derived from TopKGMM, and if we have multiple customer types, we use a mixture of several TopKGMM's where each may have different parameters.

Given a top-$k$ list $\tau \in T_{k,N}$, we define the *choice* function $\mathcal{C}_\tau : 2^{[n]} \to N$ as follows: for any assortment $\mathcal{A} \subset [n], \mathcal{C}_\tau(\mathcal{A}) = i$, iff $i$ is the highest ranked element in $\mathcal{A} \cup \{\varnothing\}$ with respect to $\tau$. If all elements of $\mathcal{A} \cup \{\varnothing\}$ are incomparable w.r.t. $\tau$, then $i$ is taken uniformly from $\mathcal{A} \cup \{\varnothing\}$.

We focus on applying the TopKGMM model to represent customer preferences. Specifically, we assume that $\tau$ is sampled from a distribution $\mathcal{D}$, where $\mathcal{D}$ is a TopKGMM distribution characterizing a single customer type. More generally, $\mathcal{D}$ can represent several customer types and we may use a mixture distribution. Since choice probabilities of a mixture distribution can simply be obtained as a linear combination of its singleton components, here we focus on singleton distributions.

## 4.1 Calculation of Choice Probabilities: DYPCHIP

Let $\mathcal{D}$ be a TopKGMM distribution on $T_{k,N}$. We let $\mathcal{C}_\mathcal{D}$ be a function mapping any assortment $\mathcal{A} \subseteq [n]$ and an option $i \in \mathcal{A} \cup \{\varnothing\}$ to the probability that $\mathcal{C}_\tau(\mathcal{A}) = i$ where $\tau$ is sampled from $\mathcal{D}$. Formally, $\mathcal{C}_\mathcal{D} : N \times 2^{[n]} \to [0, 1]$ is defined as follows:

$$\mathcal{C}_\mathcal{D}(i, \mathcal{A}) = \sum_{\tau \in T_{k,N}} \mathbb{P}_\mathcal{D}[\tau] \cdot \mathbb{1}\left(C_\tau(\mathcal{A}) = i\right) . \tag{4}$$

We now introduce DYPCHIP which calculates *choice probabilities* as defined in Equation (4), and its correctness and runtime complexity is stated in the following theorem:

**Theorem 4.1** (Calculation of choice probabilities). *Given an assortment $\mathcal{A} \subseteq [n]$ and a TopKGMM instance $\mathcal{D}$, DYPCHIP calculates $\mathcal{C}_\mathcal{D}(j, \mathcal{A})$ for all $j \in \mathcal{A} \cup \{\varnothing\}$ in $O\left(2^{\min\{k, n-k\}} k^3 |\mathcal{A}|^2\right)$.*

The main idea of this algorithm is to find these probabilities by conditioning on a given profile. Our dynamic programming tables are defined based on the order in which items are considered in PRIM.

**Overview of DYPCHIP**   Consider profile $S \subseteq [k]$ with $\ell = |S|$. Let $\mathcal{A}^{\varnothing} = \mathcal{A} \cup \{\varnothing\}$. For a top-$k$ list $\tau \in T(S)$, item $a$ from $\mathcal{A}$ is picked if (i) $\tau$ does not include any item from $\mathcal{A}^{\varnothing}$ and so $a$ is picked randomly from $\mathcal{A}^{\varnothing} \subseteq \bar{\tau}$, or (ii) $a$ is top-ranked in $\mathcal{A}^{\varnothing}$ w.r.t $\tau$. We handle the two cases separately:

1. Let $\bar{\pi}_S(a)$ denote the probability associated to case (i). Note that $\tau \cap \mathcal{A}^{\varnothing} = \emptyset$ implies that $\tau$ is from a profile $S$ with $S \cap \mathcal{A} = \emptyset$. Thus for any $a \in \mathcal{A}^{\varnothing}$:

$$\bar{\pi}_S(a) = \mathbf{P}\left(\tau \cap \mathcal{A}^{\varnothing} = \emptyset\right) \cdot \frac{1}{|\mathcal{A}| + 1} = \mathbb{1}\left(\mathcal{A}^{\varnothing} \cap S = \emptyset\right) \cdot \frac{\binom{n-k-(|\mathcal{A}|+1)}{k-\ell}}{\binom{n-k}{k-\ell}} \cdot \frac{1}{|\mathcal{A}| + 1} \ .$$

2. Let $\bar{\mathcal{A}} \doteq \mathcal{A}^{\varnothing} \cap \bar{\tau}$. The elements in $\mathcal{A}^{\varnothing}$ who have a non-zero probability of being top ranked at some iteration of DP table are only in $\bar{\mathcal{A}} \cup S$. We calculate these probabilities by conditioning on two other parameters: (1) where in PRIM algorithm they have a chance of being sampled, and (2) the position in which they are positioned in the top-$k$ list when they are the winner, i.e, ranked highest among $\mathcal{A}^{\varnothing}$. We use a three dimensional dynamic programming table $\pi_S$ as follows: Let $\{a_1, a_2, \ldots, a_r, s_\ell, s_{\ell-1}, \ldots, s_1\}$ be an ordering of the elements in $\bar{\mathcal{A}} \cup S$ where the first segment is an arbitrary ordering of $\bar{\mathcal{A}}$ and the second is the ordering of $S$ used in PRIM.
   For $q = 0$, we let $\pi_S(i, j, q)$ be the probability that any element $a_i \in \{a_1, a_2, \ldots, a_r\}$ is ranked $j$th and higher than all other elements of $\bar{\mathcal{A}}$ (see Equation (6)). Then, by iterating over $q = 1, 2, \ldots \ell$, we consider the $q$th iteration of the for loop in PRIM, and for any position $1 \leq j \leq q$, we define $\pi_S(i, j, q)$ be the probability that after completion of the $q$th iteration of the for loop in PRIM, $a_i$ is the highest element in $\mathcal{A}^{\varnothing}$ (the winner) which has so far been sampled, and $a_i$ is so far ranked $j$-th.
   We update these probabilities by conditioning whether the newly inserted element, i.e., $s_{\ell+1-q}$ is added before the prior winner, or after it. Note that, since we the profile $S$ is fixed, the probability of inserting a new element in a particular location may be obtained from Equation (3). For details of the recursive definition of $\pi_S(i, j, q)$ please see Appendix A.2. After the dynamic programming table is filled, we may calculate the choice probability of each $a_i \in \mathcal{A}^{\varnothing}$ conditioned on profile $S$ with $|S| = \ell$ by:

$$\pi_S(a_i) = \sum_{j=1}^{\ell} \pi_S(i, j, \ell) \ .$$

In each cell of the table, we need to look at $O(k)$ other cells which involves at most $O(k|\mathcal{A}|)$ operations so each table entry can be calculated in $O(k|\mathcal{A}|)$ time. Finally, we may calculate the

$$\forall a \in \mathcal{A}, \ \mathcal{C}_{\mathcal{D}}(a, \mathcal{A}) = \sum_{S \subseteq [k]} \mathbb{P}_{\mathcal{D}}[S]\left(\pi_S(a) + \bar{\pi}_S(a)\right) \ . \tag{5}$$

**Runtime of DYPCHIP**   The size of the dynamic programming table is $|\mathcal{A}| k^2$, and calculation of each element needs $k|\mathcal{A}| + n$ operations. In Equation (5) we have a sum over all profiles; which is bounded by $O(2^{\min\{k, n-k\}})$. Thus, in total all choice probabilities may be calculated in time $\Theta(2^\gamma k^3 |\mathcal{A}|^2 + 2^\gamma k^2 |\mathcal{A}| n); \gamma = \min\{k, n-k\}$.

## 5   Learning the center from choice data

In this section, we focus on the problem of learning the center of a TopKGMM distribution from *choice data*. This task is more challenging than learning from complete top-$k$ samples, as each data point provides limited information and the choice data depends on the specific assortments presented.

Consider a TopKGMM distribution $\mathcal{D}$ on $T_{k,N}$, the goal is to construct the center of $\mathcal{D}$, namely $\tau^*$ by observing choice data. The choice data consists of pairs $(\mathcal{A}_t, c_t)$, where $\mathcal{A}_t$ represents the assortment offered at round $t$, and $c_t$ is the item selected by the customer from $\mathcal{A}_t$. Formally, we let $\mathbf{D} = \langle (c_1, \mathcal{A}_1), (c_2, \mathcal{A}_2), \ldots, (c_T, \mathcal{A}_T) \rangle$; where for $t = 1, 2, \ldots, T$ we have $c_t = \mathcal{C}_\tau(\mathcal{A}_t), \tau \sim \mathcal{D}$. While in traditional learning algorithms $\mathbf{D}$ is given as input. Here, we focus of *active* learning and collect choice data by presenting assortments to the customers and recording observed choices.

Our active learning algorithm for the estimation of the center is BUCCHOI. It takes as input the set of products $N$ and assortment size $\ell$ and offers a sequence of assortments $\mathcal{A}_1, \mathcal{A}_2, \ldots \mathcal{A}_T$ to the

| **Algorithm 1** FINDTOP | **Algorithm 2** BUCCHOI |
|---|---|
| 1: **Input:** Assortment $\mathcal{A}$, a sequence $(c_1, c_2, \ldots, c_m)$ where $c_i = \mathcal{C}_\tau(\mathcal{A})$ for $\tau$ sampled from $\mathcal{D}$. | 1: **Input:** $N$: set of products, $r$: assortment size, $m$: number of samples, choice oracle $\mathcal{C}_\tau$; $\tau \sim$ TopKGMM $\mathcal{D}$. |
| 2: **Output:** $\mathcal{C}_{\tau^*}(\mathcal{A})$ if $\tau^* \cap \mathcal{A}^\varnothing \neq \emptyset$. | 2: **Output:** $k$ size of center, $\tau^*$ center of $\mathcal{D}$ |
| 3: $X \leftarrow \mathbf{0}_{\mathcal{A}^\varnothing \times \mathcal{A}^\varnothing}$ | 3: $T = \emptyset$, $B = \emptyset$, $U = N$ |
| 4: **for** $i = 1 : m$ **do** | 4: **repeat** |
| 5:     **for** $a \in \mathcal{A}^\varnothing \setminus \{c_i\}$ **do** | 5:     $\mathcal{A}$ = assortment of size $r$ from $U^a$ |
| 6:       $X_{c_i a} = X_{c_i a} + 1$ | 6:     $S = \emptyset$ # collect choice data $S$ by showing $\mathcal{A}$ repeatedly |
| 7:       $X_{a c_i} = X_{a c_i} - 1$ | 7:     **for** $j = 1 : m$ **do** |
| 8:     **end for** | 8:       $S = S \cup \mathcal{C}_\tau(\mathcal{A})$ |
| 9: **end for** | 9:     **end for** |
| 10: $Y \leftarrow X/m$ | 10:     $a = $ FINDTOP$(\mathcal{A}, S)$ |
| 11: **if** $\exists a : Y_{aa'} > \frac{1+\|\mathcal{A}\|}{2} \forall a' \in \mathcal{A}^\varnothing \setminus \{a\}$ **then** | 11:     **if** $a \neq$ None **then** |
| 12:     **Return** $a$ | 12:       $T = T \cup \{a\}$, $U = U \setminus \{a\}$ |
| 13: **else** | 13:     **else** |
| 14:     **Return** None | 14:       $B = B \cup \mathcal{A}$, $U = U \setminus \mathcal{A}$ |
| 15: **end if** | 15:     **end if** |
| | 16: **until** $U = \emptyset$ |
| | 17: $k = \|U\|$ |
| | 18: $\tau^* = $ SORTCNTR$(U, r, m)$ # (Algorithm 6) |
| | 19: **Return** $k, \tau^*$ |

---

[a] If $\|U\| < r$ use some elements from $B$

customers, recoding corresponding choices $c_t$ for $t = 1, 2, \ldots, T$. A Pseudocode of BUCCHOI is presented in Algorithm 2. Assume TopKGMM distribution $\mathcal{D}$ with center $\tau^*$, and parameters $\beta$, $p$, $\vec{w}$. Let $w_{\min} \doteq \min_{i \in k} w_i$, and $n = |N|$. The following theorem shows the sample complexity of BUCCHOI:

**Theorem 5.1.** *Assume that $\beta \geq \log 3/w_{\min}$ and $\varnothing \notin \tau^*$. By only receiving $N$ and $r$ as input and being able to collect choice samples $\mathbf{D}$ by selecting assortments, with probability at least $1 - o(1)$, we are able to learn $\tau^*$ and $k$ from $\mathbf{D}$ using only $\Theta(r^2 \log n)$ choice samples.*

The main building block of BUCCHOI is a procedure FINDTOP which given an assortment $\mathcal{A}$ and a set of choice samples $\mathbf{D}$, outputs $i \in \mathcal{A}^\varnothing$ such that $i$ has the highest rank w.r.t. $\tau^*$ among elements of $\mathcal{A}^\varnothing$. If all elements of $\mathcal{A}^\varnothing$ are incomparable in $\tau^*$, FINDTOP returns None. We note that FINDTOP in *not* an active learning algorithm and receives $\mathbf{D}$ as input. We also assume that the choice data were collected by presenting a single arbitrary assortment $\mathcal{A}$.

**FINDTOP** A pseudocode for FINDTOP is presented in Algorithm 1. The main idea is to maintain for each $i, j \in \mathcal{A}^\varnothing$, a variable $X_{ij}$. For any choice sample $c_t$ we increment $X_{ij}$ if $i$ is chosen over $j$, i.e., $i = c_t$ and $j \in \mathcal{A} \setminus \{c_t\}$, and we decrement it otherwise. Taking $Y_{ij} = X_{ij}/m$ for all $i, j \in \mathcal{A}^\varnothing$, it is not difficult to see that:

$$\mathbb{E}[Y_{ij}] = \mathcal{C}_\tau(i, \mathcal{A}) - \mathcal{C}_\tau(j, \mathcal{A}), \quad \tau \sim \mathcal{D} .$$

Based on this observation and by calculating a lower bound on $\mathcal{C}_\tau(i, \mathcal{A}) - \mathcal{C}_\tau(j, \mathcal{A})$ when $i$ is $\mathcal{A}^\varnothing$'s top element w.r.t. $\tau^*$ and $j \in \mathcal{A}^\varnothing \setminus \{i\}$ we are able to show that the top element will be found by FINDTOP if the discrepancy parameter $\beta$ is large enough and we have enough samples. Formally we show the following lemma whose proof is presented in full details in Appendix A.3:

**Lemma 5.2.** *Assume that $\beta \geq \log(3)/w_{\min}$ and let $r = |\mathcal{A}|$ and $\zeta \geq 1$ arbitrary constant. If $\mathcal{A}$ appears at least $\Theta(\zeta(r+1)^2 \log n)$ times among the displayed assortments, with probability at least $1 - o(n^{-\zeta})$ we have: if $\mathcal{A}^\varnothing \cap \tau^* \neq \emptyset$, FINDTOP will return $i \in \mathcal{A}^\varnothing$ such that $i \succ_{\tau^*} j$ for any $j \in \mathcal{A}^\varnothing \setminus \{i\}$, otherwise, it returns None, and we can conclude that $\mathcal{A}^\varnothing \cap \tau^* = \emptyset$ .*

**BUCCHOI** BUCCHOI first identifies all elements in $N$ which are ranked above $\varnothing$ and are in $\tau^*$. To this end, we maintain three sets: $T \subseteq \tau^*$ and $B \subseteq \bar{\tau}^*$, $U$ unknown whether they are in $\tau^*$ or $\bar{\tau}^*$. Assortments of size $r$ are selected repeatedly and after calling FINDTOP we either find the top

element in the assortment – which has to be in $\tau^*$ or we find out that none of the elements in the assortment are in $\tau^*$. Note that the number of times that Repeat loop iterates is bounded by $k + n/r$. Finally we call Algorithm 6 to find the rank of items in $\tau^*$. Theorem 5.1 will be concluded from Lemma 5.2 and using a union bound on all FINDTOP calls. We remark that if $\varnothing \in \tau^*$, BUCCHOI will be able to return the top prefix of $\tau^*$ constituting of the elements ranked above $\varnothing$ (see Appendix A.3).

## 6 Experiments

In this section, we present our experimental analysis, designed to achieve two main objectives: (1) to compare the predictive power of the top-$k$ Mallows model (TopKGMM) with that of the multinomial logit model (MNL), and (2) to evaluate the accuracy and computational complexity of our methods, namely PRIM sampling algorithm, the DYPCHIP choice probability computation, and the two learning algorithms, FINDTOP and BUCCHOI. The code and log files are available publicly[3]. Results are generated by running the code on a MacBook Pro M1 Max, 32GM RAM.

**Predictive power of top-$k$ MM compared to MNL: experiments on real-world data**  We used "Sushi Preference Data Set"(Kamishima et al., 2005) which contains preference of customers over a set of 100 different sushi types [4]. The data-set includes 5K preferences in the form of top-10 lists.

*Set-Up.* We begin by randomly splitting the 5K top-10 preference data into a training set (80%) and a test set (20%). We apply BUCCHOI using assortments of size one or two (Algorithm 7) to the training set using various values of $p$ and $\beta$ to learn the center of the distribution. With the learned parameters, we use DYPCHIP to compute the corresponding choice probabilities.
For evaluation, we use empirical choice probabilities on the test set by repeatedly sampling random assortments and recording corresponding choices. These empirical estimates are then compared to the predictions from DYPCHIP to assess out-of-sample accuracy; the errors are reported in Table 1. Based on these results, we identify the values of $p$ and $\beta$ that yield the lowest test error. Figure 1a shows the prediction error of the TopKGMM compared to MNL after tuning.

Considering multiple customer types, we cluster the training data (into 2–5 groups) using the Kendall's Tau distance $\mathcal{K}^p$, varying $p$. Clusters with positive silhouette scores are retained, and choice probabilities are computed within each cluster using both TopKGMM and MNL. Final predictions are weighted averages based on cluster sizes. Figure 1b shows the results for the two-clusters.

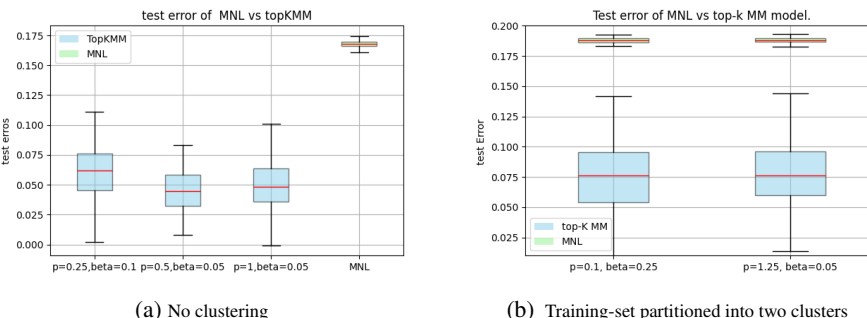

(a) No clustering    (b) Training-set partitioned into two clusters

Figure 1: Test error of the top-$k$ Mallows model compared to MNL. Parameter $\beta$ and $p$ have been selected to derive highest accuracy. Tables 1 and 2 show the test error for all choices of $p$ and $\beta$.

*Experimental Findings.* Our findings show high accuracy of out-of-sample choice probability prediction of TopKGMM compared to the accuracy obtained from MNL. These results are consistent with findings of Désir et al. (2021) who observe the same for classic Mallows model on permutations.

**Accuracy and complexity of algorithms: experiments on synthetic Data**  We use synthetic data to evaluate the accuracy and complexity of our algorithms, as it provides access to ground-truth choice probabilities and distribution centers—information unavailable in real-world datasets. This enables controlled analysis of sample complexity trade-offs with respect to key parameters: $n$ (number of products), $k$ (size of top-$k$), $r$ (assortment size), $\beta$ (decay parameter) and $p$ (Kendall's Tau parameter).

---

[3]Link to the code `https://github.com/ShahrzadGit/topkmallows-choices`
[4]Link to of Sushi Preference Data Set `https://www.kamishima.net/sushi/`.

*Accuracy and time complexity of* PRIM *and* DYPCHIP. We evaluate algorithm accuracy by generating samples from a TopKGMM using PRIM and comparing empirical choice frequencies over random assortments to the probabilities predicted by DYPCHIP. This is repeated across 20 assortments, with the mean and standard deviation of results shown in Figure 2a. The run-times of DYPCHIP and PRIM are reported in Appendix B.2.

*Accuracy and sample complexity of learning algorithms of* BUCCHOI *and* FINDTOP. We evaluate our two learning algorithms by generating $m$ samples from TopKGMM distribution using PRIM and running FINDTOP and BUCCHOI to learn the top element or distribution center. We assess the convergence of these methods by comparing the learned and true values. When learning the center in BUCCHOI, we use the Kendall's Tau distance $\mathcal{K}^p$ of learned and true center. In FINDTOP we check whether the learned top element is the same as the true top element and directly calculate accuracy based on the frequency of matching values. Each experiments is repeated 10 times across a range of model parameters and average and standard deviation are obtained; see Figure 2, and Appendix B.3.

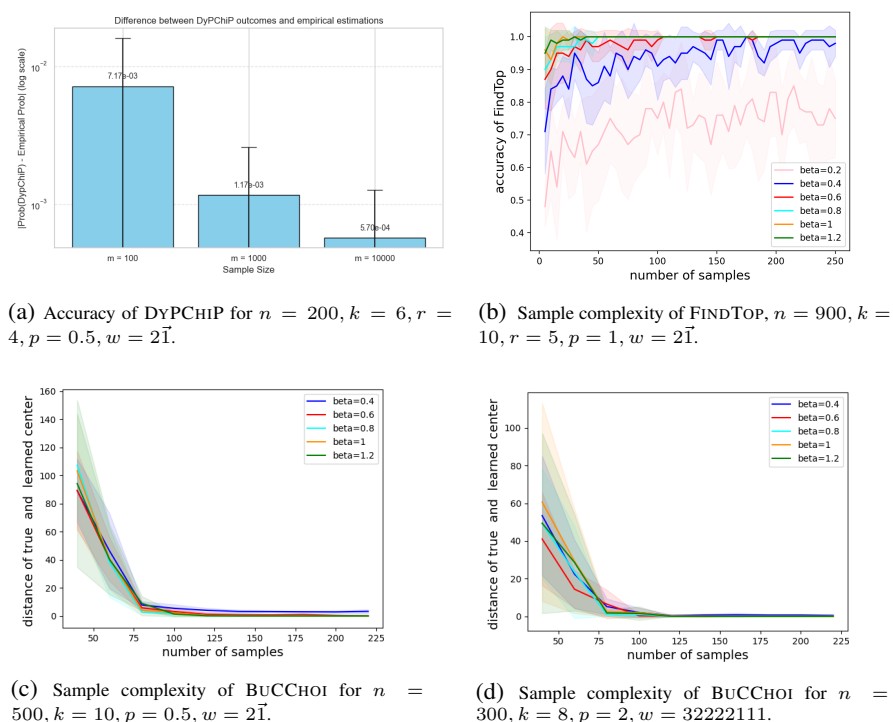

(a) Accuracy of DYPCHIP for $n = 200, k = 6, r = 4, p = 0.5, w = 2\vec{1}$.

(b) Sample complexity of FINDTOP, $n = 900, k = 10, r = 5, p = 1, w = 2\vec{1}$.

(c) Sample complexity of BUCCHOI for $n = 500, k = 10, p = 0.5, w = 2\vec{1}$.

(d) Sample complexity of BUCCHOI for $n = 300, k = 8, p = 2, w = 32222111$.

Figure 2: performance of algorithms DYPCHIP, FINDTOP and BUCCHOI on synthetic data.

*Experimental Findings.* Our experiments in this section support the theoretical results on the accuracy and complexity of our algorithms. Notably, we observe that our methods achieve high accuracy with a relatively small number of samples—often logarithmic in the number of items $n$.

For DYPCHIP, we find that time complexity increases rapidly with $k$, as expected due to its exponential dependence on this parameter (See Figure 3). In contrast, the runtime shows minimal sensitivity to the number of products $n$. For PRIM, the exponential dependence on $k$ is less restrictive since it primarily affects the preprocessing step; once this step is completed, generating a large number of samples remains efficient with low amortized cost (See Table 3).

In the learning algorithms BUCCHOI and FINDTOP, we observe that the sample complexity increases as $\beta$ decreases. This is expected, as smaller values of $\beta$ cause the distribution to approach uniformity, reducing the concentration of samples around the center and making learning more challenging.

## 7 Conclusion

In conclusion, the generalized Mallows model for top-$k$ lists provides a more realistic framework for understanding user preferences, particularly when users care only about a limited set of options. Our work centers on applying TopKGMM to choice modeling and developing several key algorithms. An important open direction is learning the model parameters for mixture versions of these models.

# 8 Acknowledgment

We are thankful to Anonymous NeurIPS reviewers for helping us strengthen the presentation of our paper. Shahrzad Haddadan is supported by Rutgers Business School's Dean's Research Seed Fund.

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

# A  Missing proofs and details

## A.1  Missing details from Section 3.2

In this section, we prove Theorem 3.1. To this end, we first establish the correctness of Lemma 3.3 that proves profile $S$ can be sampled according to $\mathbb{P}_\mathcal{D}[S]$.

**Lemma** (Restatement of Lemma 3.3). *Given a TopKGMM distribution $\mathcal{D}$ with parameters $\beta$, $p$ and $w$, any profile $S = \{s_1, s_2, \ldots, s_\ell\} \subseteq [k]$ ($s_1 < s_2 < \cdots < s_l$) has probability $\mathbb{P}_\mathcal{D}[S]$ proportional to $\exp\left(-\beta f(S)\right) Z(S)$, where:*

$$f(S) \quad := \quad w_0 pQ(S) + \sum_{j \in [k] \setminus S} w_j (I_j(S) + pP_j(S))$$

$$Z(S) \quad := \quad \binom{n-k}{k-\ell}(k-\ell)! \prod_{j=1}^{\ell} \sum_{r=0}^{k-j} e^{-\beta w_{s_j} r}$$

*and it can be sampled in $O(2^k k)$.*

*Proof.* Let $\mathcal{M}$ be the normalizing factor in Equation (TopKGMM), so $\mathcal{M} = \sum_{\tau \in T_{k,N}} \mathbb{P}_\mathcal{D}[\tau]$. Then

$$\mathbb{P}_\mathcal{D}[S] = \frac{1}{\mathcal{M}} \sum_{\tau \in T(S)} \mathbb{P}_\mathcal{D}[\tau] \hspace{4cm} \text{(Equation (2))}$$

$$= \frac{1}{\mathcal{M}} \sum_{\tau \in T(S)} \exp\left(-\beta(w_0 pQ(\tau) + \sum_{j \in [k]} w_j \cdot (I_j(\tau) + pP_j(\tau)))\right)$$

$$\text{(Equation (TopKGMM))}$$

$$= \frac{1}{\mathcal{M}} \sum_{\tau \in T(S)} \exp\left(-\beta(w_0 pQ(S) + \sum_{i \in [k] \setminus S} w_j \cdot (I_j(S) + pP_j(S)) + \sum_{j \in S} w_j \cdot I_j(\tau))\right)$$

$$= \frac{1}{\mathcal{M}} \sum_{\tau \in T(S)} \exp\left(-\beta(w_0 pQ(S) + \sum_{i \in [k] \setminus S} w_j \cdot (I_j(S) + pP_j(S)))\right) \cdot \exp\left(-\beta \sum_{j \in S} w_j \cdot I_j(\tau)\right)$$

$$= \frac{\exp(-\beta f(S))}{\mathcal{M}} \sum_{\tau \in T(S)} \exp\left(-\beta \sum_{j \in S} w_j \cdot I_j(\tau)\right)$$

The remaining sum relates to the inversions of elements in $S$. Based on Definition 2.1, an element $s_j \in S \subseteq [k]$ makes an inversion with another element $s$ iff $s_j \succ_{\tau^*} s$ and $s \succ_\tau s_j$. Note that since $\tau \in T_{\tau^*}(S)$, the possibilities for such $s$ are $\{s_{j+1}, \cdots, s_\ell\}$ and the elements of $[n] \setminus [k]$ which are now in $\tau$, i.e., $\tau \cap \{k+1, \cdots, n\}$. Note that $\tau \cap \{k+1, \cdots, n\} = k - \ell$. Thus, the inversion of $j$ can be any number between 0 to $(\ell - j) + (k - \ell)$ which is any number from 0 to $k - j$. Furthermore, for any valid selection of these values for elements $j \in S$ w.r.t. defined ranges, then the position of these elements in $\tau$ are uniquely determined. This can be achieved by starting with a sequence of $k - \ell$ stars which would correspond to any selection of elements from $\{k+1, \cdots, n\}$, and then inserting element $s \in (s_\ell, s_{\ell-1}, \cdots, s_1)$ iteratively based on their value of $I_s$. This shows the 1:1 correspondence between values of $I_j$s and how elements of $S$ are positioned in $\tau$. Since there are $\binom{n-k}{k-l}(k-l)!$ possible cases for the arrangement of the remaining elements, we get that

$$\sum_{\tau \in T(S)} \exp\left(-\beta \sum_{j \in S} w_j \cdot I_j(\tau)\right) = \binom{n-k}{k-l}(k-l)! \sum_{\substack{\text{valid choice of } I_s \\ \text{for all } s \in S}} \exp(-\beta \sum_{j \in S} w_j \cdot I_j) = Z(S).$$

where the last equality follows from the fact that depending on the value of $I_{s_j}$, any of the terms in the summation $1 + e^{-w_{s_j}\beta} + \cdots + e^{-(k-j)w_{s_j}\beta}$ appears once. Taking product over the terms selected from these sums would exactly correspond to a valid selection of $I_s$ for all $s \in S$. Since $\mathcal{M} =$

$\sum_{\tau \in T_{k,N}} \mathbb{P}_{\mathcal{D}}[\tau]$ and each $\mathbb{P}_{\mathcal{D}}[\tau]$ contributes to the $\mathbb{P}_{\mathcal{D}}[S]$, the defined values form a probability distribution over different subsets of $[k]$ where $\mathbb{P}_{\mathcal{D}}[S]$ is proportional to $\exp(-\beta f(S)) Z(S)$.

It remains to show that this value can be computed in $O(2^\gamma k)$; $\gamma = \min\{k, n-k\}$. We argue that for each $S$, $Z(S)$ and $f(S)$ can be calculated in $O(k)$, the total number of profiles is bounded by $2^\gamma$, hence all the computation can be done in $O(2^\gamma k)$. For $Z(S)$, the coefficient before sum has at most $2k$ terms and there are at most $k$ terms in the sum where each term corresponds to a geometric series, so $Z(S)$ can easily be calculated in $O(k)$. $f(S)$ can be calculated in $O(k)$ using the following algorithm which clearly has $O(k)$ runtime.

---

**Algorithm 3** TOPKGMMSAMPLING (TOPKGMM)

---

**Input:** TopKGMM $\mathcal{D}(\beta, \boldsymbol{w})$.
**Output:** Sampled Top-$k$ ordering $\tau$ proportional to $\mathcal{D}$
$f(S) = $ PROFILE PROBABILITY
Sample $S$ proportional to $\mathbb{P}_{\mathcal{D}}[S] = f(S) \cdot Z(S)$.
**Return** PRIM$(\beta, \boldsymbol{w}, S)$.

---

---

**Algorithm 4** PROFILE-BASED RIM(PRIM)

---

**Input:** TopKGMM$(\beta, \boldsymbol{w})$, $S = \{s_1, s_2, \ldots, s_\ell\} \subseteq [k]$ where $s_1 < s_2 < \cdots < s_\ell$.
**Output:** Top-$k$ list $\tau \in T_\sigma(S)$.
$A \leftarrow$ ordered random $k - \ell$ elements from $[n] \setminus [k]$
**for** $s \leftarrow s_\ell, s_{\ell-1}, \cdots, s_1$ **do**
 Insert $s$ at position $j$ in $A$ w.p. $\frac{\exp(-\beta w_s j)}{\sum_{x=0}^{|A|} \exp(-\beta w_s x)}$ .
**end for**
**Return** $\tau = A$

---

---

**Algorithm 5** PROFILE PROBABILITY

---

**Input:** Profile $S = \{s_1, s_2, \ldots, s_\ell\} \subseteq [k]$.
**Output:** $f(S)$
$Q(S) \leftarrow \binom{k-\ell}{2}$.
$x, y \leftarrow 0, 0$
**for** $j \in \{k, k-1, \cdots, 1\}$ **do**
 $I_j \leftarrow k - \ell + x$
 $P_j \leftarrow n - 2k + \ell + y$
 **if** $j \in S$ **then**
  $x \leftarrow x + 1$
 **else**
  $y \leftarrow y + 1$
 **end if**
**end for**
**Return** $f(S) := w_0 p Q(S) + \sum_{j \in [k] \setminus S} w_j (I_j(S) + p P_j(S))$.

---

$\square$

Using Lemma 3.3 and analyzing Algorithm 3, we can prove Theorem 3.1.

***Proof of Theorem 3.1.*** Any $\tau \in T_{k,N}$ with $S = \tau \cap [k]$, according to Algorithm 4, is sampled with probability

$$\frac{1}{\binom{n-k}{k-l}(k-l)!} \cdot \frac{\sum_{j \in S} \exp(-\beta w_j I_j)}{\sum_{j=1}^{l} \prod_{r=0}^{k-j} \exp(-\beta w_{s_j} r)} = \frac{\sum_{j \in S} \exp(-\beta w_j I_j)}{Z(S)}.$$

Since each $S$, is sampled by $\frac{\exp(-\beta f(S)) Z(S)}{\sum_{S \subseteq [k]} \exp(-\beta f(S)) Z(S)}$, we get that $\tau$ is sampled with

$$\frac{\exp(-\beta f(S)) Z(S)}{\sum_{S \subseteq [k]} \exp(-\beta f(S)) Z(S)} \cdot \frac{\exp(\sum_{j \in S} -\beta w_j I_j)}{Z(S)} = \frac{\exp(-\beta(f(S) + \sum_{j \in S} -w_j I_j))}{\sum_{S \subseteq [k]} \exp(-\beta f(S)) Z(S)}$$

where the denominator as previously discussed in proof of Lemma 3.3, is equal to normalizing factor $\mathcal{M} = \sum_{\tau \in T_{k,N}} \mathbb{P}_{\mathcal{D}}[\tau]$. Since numerator is just restating Equation (TopKGMM), each $\tau$ is sampled w.r.t Equation (TopKGMM).

**Time complexity** An upperbound on the number of profiles is $2^\gamma; \gamma = \min\{k, n-k\}$. Therefore $Z(S)$ and $f(S)$ can be computed in $O(2^\gamma k)$ by Lemma 3.3, and we can sample $S$ in $O(2^\gamma k)$. Next, fixing $S$, we need to execute Algorithm 4, where we first need to sample $k - \ell$ elements which an be done in $O(k \log n)$. And then inserting elements of $S$ with respect to probabilities that can be calculated in $O(k)$ for each element of $S$, so requiring $O(k^2)$ time for the recursive insertion loop. Hence overall the algorithm runs in $O(2^\gamma k + k \log n + k^2); \gamma = \min\{k, n-k\}$. $\qquad\square$

## A.2 Missing details from Section 4.1

In this section, we provide the details in design of DYPCHIP. The main remaining part is the calculation of $\pi_S(i, j, q)$ for given $\mathcal{A}, i = 1, \ldots r, r+1, \ldots r+\ell, q = 0, 1, \ldots \ell$, and $j = r+1, r+2, \ldots r+q, k$; where $r = |\bar{\mathcal{A}}|$ and $\ell = |S|$. We remind the reader that in $\pi(i, j, q)$:

- $i$ indicated the index of an item $a$ whose choice probability is being calculated. We use $L$ to refer to the ordered set $\{a_1, a_2, \ldots, a_r, s_\ell, s_{\ell-1}, \ldots, s_1\}$ a ranking of the elements in $\bar{\mathcal{A}} \cup S$ where $a_1, a_2, \ldots, a_r$ is an arbitrary ranking of $\bar{\mathcal{A}}$ and $s_\ell, s_{\ell-1}, \ldots, s_1$ is the ranking of $S$ used in PRIM. Thus, when we say $a_i$ we mean the $i$th element in $L$.

- $j$ indicates the position of $a_i$ if it is the winner until the $q$th iteration of the for loop in PRIM. When $j = k$ we mean that the winner is in $\bar{\tau}$.

- $q$ denote that so far we are considering only elements that have been sampled until the $q$th iteration of the for loop in PRIM.

Since $\pi_S(i, j, q)$ can only take non-zero value when $a_i$ is sampled in the top-$k$ list, we either have $a_i \in \bar{\tau}$ or $a_i \in S$. We analyze each of these two possible cases separately. We first initialize the DP table for $q = 0$ by considering $i = 1, 2, \ldots r$. Then we consider items that are in $S$ in the reverse order of their priority (consistent with the Algorithm 4).

1. Let $\bar{\mathcal{A}} = \mathcal{A}^\varnothing \cap \bar{\tau}$. Before start of the loop in PRIM, only elements of $\bar{\mathcal{A}}$ have been sampled and have nonzero probability of being selected. These elements are the first $r$ elements in our list. Thus, for $i = 1, 2, \ldots, r$, we store these probabilities in the DP table as:

$$
\begin{cases}
\pi(i, j, 0) = \underbrace{\frac{1}{n-k-j+1}}_{\substack{\text{the probability that} \\ i \text{ is sampled at position } j}} \underbrace{\prod_{j'=1}^{j-1} (1 - \frac{r}{n-k-j'})}_{\substack{\text{the probability that} \\ \text{no element of } \bar{\mathcal{A}} \text{ is sampled} \\ \text{in positions } j-1}} & \text{for } j \leq r \\
\pi(i, k, 0) = \frac{\binom{n-k-(r+1)}{k-\ell}}{\binom{n-k}{k-\ell}} \cdot \frac{1}{r+1} & \text{the case that } \bar{\mathcal{A}} \subseteq \bar{\tau}
\end{cases}
\tag{6}
$$

2. For $q = 1, 2, \ldots, \ell$, consider the for loop in PRIM and let $a_{cur}$ be the element that is inserted in the top-$k$ list at iteration $q$ of this loop. Therefore, $a_j = s_{\ell+1-q}$, and in our list it is ranked $(r+q)$th, thus, $cur = r + q$. We distinguish between the two cases where $a_{cur} \in \mathcal{A}^\varnothing$ or not.

   *Case 1.* $a_{cur} \notin \mathcal{A}^\varnothing$: since $a_{cur}$ is not a choice option, we have that $\pi_S(cur, j, q) = 0$ for all $j \leq q$. For any $i < cur$, if $a_{cur}$ is inserted higher than the previous winner, the winner position will increment. If $a_{cur}$ is inserted lower than the previous winner, the winner position will not change. Therefore $\forall i < cur$,

   $$\pi_S(i, j, q) = \pi_S(i, j, q-1) \cdot \text{PRIMPOS}(S, > j, q) + \pi_S(i, j-1, q-1) \cdot \text{PRIMPOS}(S, \leq j-1, q),$$

   where $\text{PRIMPOS}(S, > j, q)$ is the probability that at iteration $q$ of PRIM we insert an element in a position after (lower than) $j$, and $\text{PRIMPOS}(S, \leq j, q)$ is it is inserted before (higher than) $j$. Note that $\text{PRIMPOS}(S, \leq j, q) = 1 - \text{PRIMPOS}(S, > j, q)$, and the probability $\text{PRIMPOS}(S, \leq j, q)$ can be directly calculated using Equation (3) by summing over all insertion probabilities for $j' = 1, 2, \ldots, j$.

*Case 2.* $a_{cur} \in \mathcal{A}^\varnothing$: In this case, if $a_{cur}$ is positioned higher than the previous winner, it will become the new winner. Otherwise, the position of the previous winner will increment. Thus,

$$\forall i < cur, \quad \pi_S(i, j, q) = \pi_S(i, j, q - 1) \cdot \text{PRIMPOS}(S, > j, q) .$$

Furthermore,

$$\pi_S(cur, j, q) = \text{PRIMPOS}(S, j, q) \cdot \sum_{i < cur} \sum_{j'=j}^{\ell} \pi_S(i, j', q - 1) ,$$

where $\text{PRIMPOS}(S, j, q)$ is the probability that at the $q$th iteration of PRIM we insert an element in location $j$ which again can be directly calculated from Equation (3).

*Proof of Theorem 4.1.* The above analysis forms a proof for the correctness of DYPCHIP. □

## A.3 Missing proofs and details from Section 5

In this section we provide the missing proof for correctness and sample complexity of BUCCHOI and FINDTOP and we present the missing details.

Let us first present the pseudocode of SORTCNTR which is called in BUCCHOI.

---
**Algorithm 6** SORTCNTR
---
**Input:**
choice oracle $\mathcal{C}_\tau$; $\tau \sim \mathcal{D}$ where $\mathcal{D}$ is a topKGMM with center $\tau^*$
$U$ containing elements of $\tau^*$ as an (unsorted) set,
$r$ assortment size
$m$ sample size
$B$ set of elements out of center
**Output:** $\tau^*$ sorted
$k = |U|$
**for** $i = 1 : k$ **do**
  # find the top element in $U$ and delete it
  $T = U$
  **repeat**
    # $T$ maintains potential top elements of $U$
    Divide $T$ to non-intersecting assortments of size $r$: $\mathcal{A}_1, \mathcal{A}_2, \ldots \mathcal{A}_\gamma$; $\gamma = \lceil i/r \rceil$ # use elements of $B$ for $\mathcal{A}_\gamma$ if needed
    **for** $\kappa = 1 : \gamma$ **do**
      $T = \emptyset$
      # collect choice samples by showing $\mathcal{A}_\kappa$ to customers and find the top element of $\mathcal{A}_\kappa$
      $S = \emptyset$
      **for** $j = 1 : m$ **do**
        $S = S \cup \mathcal{C}_\tau(\mathcal{A}_\kappa)$
      **end for**
      $T = T \cup \{\text{FINDTOP}(\mathcal{A}_\kappa, S)\}$
    **end for**
  **until** $|T| = 1$
  $U = U \setminus T$
**end for**
**Return** $\tau^*$
---

**SORTCNTR** When BUCCHOI finds all the elements that are in the center, we call SORTCNTR to find the ranking of these items as a list. In the for loop the item which is positioned the $i$th will be found and deleted from $U$. In other words, we first find the top element of $U$ this is the element ranked first, after deleting it, we find the top element in the remainder which is ranked second and so on.

In order to find the top element we use a Repeat-Until loop. In this loop $U$ is divided to assortments of size $r$ and using FINDTOP we find the winner in each assortment and continue by having a tournament between the winners until the winner of all (which is the top element) is found.

Note that the repeat-until loop iterates at most $\log_r(k)$ times. We call FINDTOP at each iteration $\lceil k/r \rceil$ times, and since we have for loop the total number of times that FINDTOP is called is bounded by $\Theta(k^2 \log_r(k))$.

We now present the proof lemmas and theorems. As before, without loss of generality we assume that $\tau^* = 1 \succ 2 \succ \cdots k \succ \{k+1, \cdots, n\}$.

**Lemma A.1.** *Let* $i < j \in \mathcal{A}_t$, *and assume* $\mathcal{I} = \{i_1, i_2, \ldots, i_\ell\} \subseteq \mathcal{A}_t$ *is such that* $i < i_\kappa < j$ *for all* $i_\kappa \in \mathcal{I}$. *We have:*

$$\mathcal{C}_{\mathcal{D}}(i, \mathcal{A}_t) \geq \mathcal{C}_{\mathcal{D}}(j, \mathcal{A}_t) \cdot \exp \left( \beta (w_i + \sum_{\kappa=1}^{\ell} w_{i_\kappa}) \right) \ .$$

Consider now a fixed $\mathcal{A}$ and let $a_r, a_{r-1}, \ldots, a_1, \mathcal{A}_0$ be such that, for each $a_i$, there are $i$ elements in $\mathcal{A}^{\varnothing}$ which are ranked under it by $\tau^*$, i.e., $|\{a \in \mathcal{A}^{\varnothing} \mid a > a_i\}| = i$. As a consequence of this definition we have: $|\mathcal{A}| = r$ and $\mathcal{A}^{\varnothing} \cap \tau^* = \{a_r, a_{r-1}, \ldots, a_1\}$, and $\mathcal{A}_0 \subseteq \bar{\tau}$. We denote any arbitrary element of $\mathcal{A}_0$ by $a_0$.

**Lemma A.2.** *For any* $\kappa = r - 1, r - 2, \ldots 1, 0$, *we have:*

$$\mathcal{C}(a_r, \mathcal{A}) - \mathcal{C}(a_\kappa, \mathcal{A}) \geq \frac{1 - \exp(-\beta w_{a_r})}{1 + r \exp(-\beta w_{a_r})} \ .$$

*Proof.* From Lemma A.1 we can immediately conclude that: for any $\kappa = r - 1, r - 2, \ldots 1, 0$, $\mathcal{C}(a_r, \mathcal{A}) \cdot \exp(-\beta w_{a_r}) \geq \mathcal{C}(a_\kappa, \mathcal{A})$.

Let us now rearrange this inequality:

$$\mathcal{C}(a_r, \mathcal{A}) \cdot \exp(-\beta w_{a_r}) \geq \mathcal{C}(a_\kappa, \mathcal{A})$$
$$\mathcal{C}(a_r, \mathcal{A}) \cdot \exp(-\beta w_{a_r}) - \mathcal{C}(a_r, \mathcal{A}) \geq \mathcal{C}(a_\kappa, \mathcal{A}) - \mathcal{C}(a_r, \mathcal{A})$$
$$\mathcal{C}(a_r, \mathcal{A})[1 - \exp(-\beta w_{a_r})] \leq \mathcal{C}(a_r, \mathcal{A}) - \mathcal{C}(a_\kappa, \mathcal{A}) \tag{7}$$

On the other hand we have that: $\sum_{\kappa=1}^{r} \mathcal{C}(a_\kappa, \mathcal{A}) + \sum_{a_0 \in \mathcal{A}_0} \mathcal{C}(a_0, \mathcal{A}) = 1$. Thus,

$$1 \leq \mathcal{C}(a_r, \mathcal{A}) + r\mathcal{C}(a_r, \mathcal{A}) \cdot \exp(-\beta w_{a_r})$$
$$\mathcal{C}(a_r, \mathcal{A}) \geq 1/(1 + r \exp(-\beta w_{a_r})) \ .$$

We now substitute this lower bound in Equation (7) and obtain the premise. $\qquad \square$

Consider now for any for any $i, j \in \mathcal{A}^{\varnothing}$, for $t = 1, 2, \cdots, T$, assume we keep displaying assortment $\mathcal{A}$ and consider the following random variable:

$$X_{ij}^t = \begin{cases} +1 & \text{if } i \text{ is chosen} \\ -1 & \text{if } j \text{ is chosen} \\ 0 & \text{otherwise} \end{cases}$$

We denote $Y_{ij} = \sum_{t=1}^{T} X_{ij}^t / T$, we have that:

$$\mathbb{E}[Y_{ij}] = \mathcal{C}(i, \mathcal{A}) - \mathcal{C}(j, \mathcal{A}) \ .$$

In the following lemma we let $w_{\min}$ be the minium weight assigned to the elements in $\tau^*$, i.e., $w_{\min} \doteq \min_{i \in k} w_i$ .

**Lemma A.3** (Restatement of Lemma 5.2). *Assume that $w_{\min} \geq \log(3)/\beta$ and let $r = |\mathcal{A}|$ and $\zeta \geq 1$ arbitrary constant. If $\mathcal{A}$ appears at least $\Theta(\zeta(r+1)^2 \log n)$ times among the displayed assortments, with probability at least $1 - o(n^{-\zeta})$ we have: if $\mathcal{A}^{\varnothing} \cap \tau^* \neq \emptyset$ we are able to identify $i \in \mathcal{A}^{\varnothing}$ such that $i < j$ for any $j \in \mathcal{A}^{\varnothing} \setminus \{i\}$, otherwise, we can conclude that $\mathcal{A}^{\varnothing} \cap \tau^* = \emptyset$.*

*Proof.* Note that if $\mathcal{A}^{\varnothing} \cap \tau^* \neq \emptyset$ there is at least one element with rank $r$ in $\mathcal{A}^{\varnothing}$. For this element we may apply Lemma A.2. Note that this element is unique. If $\mathcal{A}^{\varnothing} \cap \tau^* = \emptyset$, then the choice probability of all the elements in $\mathcal{A}^{\varnothing}$ are equal to $1/(r+1)$.

Assuming $w_{\min} \geq \log 3/\beta$, we may bound the right-hand side of Lemma A.2 as follows:

Since $a_r \in \tau^*$

$$1 - \exp(-\beta w_{a_r}) \geq 1 - \exp(-\beta w_{\min}) \geq 1 - \exp(\log 3)$$
$$\geq 2/3 .$$

Therefore,

$$\frac{1 - \exp(\beta w_{a_r})}{1 + r \exp(\beta w_{a_r})} = \frac{1 - \exp(-\beta w_{a_r})}{1 - r(1 - \exp(-\beta w_{a_r})) + r}$$

We substitute $x = 1 - \exp(-\beta w_{a_r})$, note $1 \geq x \geq 2/3$.

$$\frac{1 - \exp(-\beta w_{a_r})}{1 + r \exp(-\beta w_{a_r})} = \frac{x}{1 + r - rx}$$

Note that we have: $\frac{1+r}{1+2r} \leq 2/3$ for $r \geq 1$.

Thus,

$$\frac{x}{1 + r - rx} \geq \frac{1}{1 + r} \iff x(1 + r) \geq 1 + r - rx$$
$$\iff x + rx \geq 1 + r - rx$$
$$\iff x(1 + 2r) \geq 1 + r$$
$$\iff x \geq (1 + r)/(1 + 2r)$$
$$\Leftarrow x \geq 2/3 .$$

We now consider random variables $Y_{ij}$ for any $i, j = 1, 2, \ldots, n$ as defined before.

We have:

$$\mathbb{E}[Y_{ij}] \begin{cases} \geq 1/(r+1) & \text{if } i = a_r \\ = 0 & \text{if } i, j \in \bar{\tau}^* \end{cases} \tag{8}$$

Using Hoeffding bound, we have:

$$\mathbb{P}\left(|Y_{ij} - \mathbb{E}[Y_{ij}]| \geq 1/2(1+r)\right) \leq 2\exp\left(-\frac{T}{32(1+r)^2}\right)$$

Which for $T \geq 64\zeta(r+1)^2 \log(n+r)$ is bounded by:

$$2\exp\left(-\frac{64\zeta(r+1)^2 \log(n+r)}{32(1+r)^2}\right) = 2\exp\left(-2\zeta \log(n+r)\right)$$
$$= 2n^{-2\zeta} \cdot r^{-2\zeta} .$$

In order to find $a_r$, we estimate $Y_{ij}$ for all $\Theta(r^2)$ pairs of $ij$ in $\mathcal{A}^\varnothing$. Using the above bound, and applying a union bound over all pairs we have:

With probability $1 - 2n^{-2\zeta}$:

$$\forall i, j \in \mathcal{A}^\varnothing, \ |Y_{ij} - \mathbb{E}[Y_{ij}]| \leq 1/2(1+r)$$

Using Equation (8), this means that:

With probability $1 - 2n^{-2\zeta}$:

$$\forall j \in \mathcal{A}^\varnothing, \ Y_{a_r j} > 1/2(1+r), \text{ and}$$
$$\forall i, j \in \bar{\tau}^*, Y_{ij} < 1/2(1+r)$$

We use the above rule to find $a_r$: first we find all $Y_{ij}$s, and output the $i$ for which we have:

$$\forall j \in \mathcal{A}^\varnothing \setminus \{i\}, \ Y_{ij} > 1/2(1+r) \ . \tag{9}$$

Note that if $a_r$ exists, there will be one unique $i \in \mathcal{A}^\varnothing$ Equation (9) holds. Therefore, we take $a_r = i$. Alternatively, if there is no element with rank $r$ we have $\forall i, j \in \mathcal{A}^\varnothing, i, j \in \bar{\tau}^*$ thus, $Y_{ij} < 1/2(1+r)$. $\square$

**Theorem A.4** (Restatement of Theorem 5.1). *Assume that $\beta \geq \log 3/w_{\min}$ and $\varnothing \notin \tau^*$. By only receiving $N$ and $r$ as input and being able to collect choice samples $\mathbf{D}$ adaptively, with probability at least $1 - o(1)$, BUCCHOI is able to learn $\tau^*$ and $k$ from $\mathbf{D}$ using only $\Theta(r^2 \log n)$ choice samples.*

*Proof.* Note that BUCCHOI calls FINDTOP at most $\Theta(k + n/r + k^2 \log_r k) = \Theta(n/r + k^2 \log_r k)$ times. We take $\zeta = 3$, and using Lemma 5.2 we have that by $\Theta\left(3(r+1)^2 \log n\right)$ choice samples which are obtained from presentation of $\mathcal{A}$ with probability $1 - o(n^{-3})$ we will be able to identity the top element of $\mathcal{A}$. Since the total number of pairs $i, j$ is $\Theta(n/r + k^2 \log k) = (n^2 \log n)$, we need to take a union bound over $n^2 \log n$ variables thus, in total the probability of failure is bounded by $o(n^2 n^{-3}) = o(1)$, from which we conclude the premise.

$\square$

# B Experimental setup and extra tables and figures

## B.1 Details of learning choice probabilities on the Sushi data-set.

As explained in the main body, we have divided the data set into two non-intersecting parts for training (80%) and test sets (20%). We use the training set to learn choice probabilities. First we employ BUCCHOI-II (Algorithm 7). BUCCHOI-II uses the top-10 sushi types as samples generated from an unknown distribution and collects choice data from it by actively selecting assortments. Since the result of BUCCHOI-II may produce partial orders, we break the ties, by using a score function based on the random variables $X_{ij}$ that are used in FINDTOP. In particular, since $X_{ij}$ shows the number of times item $i$ beats $j$ we use $\sum_{j \in N} X_{ij}$ as a tie breaking score for item $i$.

This is needed because we need to feed the learned center from BUCCHOI-II to DYPCHIP to learn choice probabilities and DYPCHIP receives as input a top-$k$ list. Note that $k$ is unknown to the algorithm and we let BUCCHOI learn it. If $k$ is too large (>15) we truncate the learned center to reduce time complexity.

We tune parameters $\beta$ and $p$ by performing a grid search over a range of values as in Table 1. From this table we conclude that $p = 0.5$ and $\beta = 0.05$ have the best test error. Thus, we select them as the model parameters. The mean and std of test error of the best parameters are reported in Figure 1a.

We handle learning mixture models by clustering the data into several clusters. First we use $\mathcal{K}^p$ as a distance metric taking $p \in [0.01, 0.25, 0.05, 0.75, 0.1, 0.25, 0.5, 1, 1.5, 2.2.5, 5]$ and then we divide data set to $2, 3$ or $5$ clusters. Most of the clusters we obtain have a negative silhouette coefficient, and the only positive ones are:

**Algorithm 7** BUCCHOI-II

---

**Input**, $N$: set of products, $m$: number of samples, choice oracle $\mathcal{C}_\tau$; $\tau \sim \mathcal{D}$ where $\mathcal{D}$ is a topKGMM.
**Output** center of $\mathcal{D}$
$T = \emptyset$
**for** $i \in N$ **do**
   $S = \emptyset$
   $\mathcal{A} = \{i\}$
   **for** $j = 1 : m$ **do**
      $c_j = \mathcal{C}_\tau(\mathcal{A})$
      $S = S \cup \{c_j\}$
   **end for**
   $T = T \cup \text{FINDTOP}(\mathcal{A}, S)$
**end for**
**for** $i, j \in T$ **do**
   $\mathcal{A} = \{i, j\}$
   **for** $j = 1 : m$ **do**
      $c_j = \mathcal{C}_\tau(\mathcal{A})$
      $S = S \cup \{c_j\}$
   **end for**
   **if** $i == \text{FINDTOP}(\mathcal{A}, S)$ **then**
      $\sigma[i] > \sigma[j]$
   **else**
      $\sigma[i] < \sigma[j]$
   **end if**
**end for**
**Return** $\sigma$

---

p= 0.1 num clusters = 2: silhouette score = 0.006287393358176328
p= 1.25 num clusters = 2: silhouette score = 0.007796245381136862
p= 2.5 num clusters = 2: silhouette score = 0.004498089502999035
p= 5 num clusters= 2: silhouette score = 0.011002010781645042
p= 5 num clusters = 3: silhouette score = 0.0023222431733978116

As we can see, even when the silhouette scores are positive they are pretty low, which suggests that one cluster is the best choice. This is consistent with the test error we obtained, as the model trained without clustering has a lower test error compared to the model trained on two clusters or more; see Figure 1b.

| $p$ \ $\beta$ | 0.05 | 0.1 | 0.25 | 0.5 | 0.75 | 1 | 1.25 | 1.5 | 1.75 | 2 |
|---|---|---|---|---|---|---|---|---|---|---|
| 0.01 | 0.1162 | 0.1047 | 0.0888 | 0.0879 | 0.0925 | 0.0979 | 0.1030 | 0.1087 | 0.1145 | 0.1202 |
| 0.025 | 0.1133 | 0.1007 | 0.0858 | 0.0872 | 0.0922 | 0.09768 | 0.1028 | 0.1085 | 0.1144 | 0.1201 |
| 0.05 | 0.1083 | 0.0938 | 0.0817 | 0.0852 | 0.0907 | 0.0964 | 0.1017 | 0.1075 | 0.1135 | 0.1193 |
| 0.075 | 0.1032 | 0.0868 | 0.0797 | 0.0841 | 0.0898 | 0.0956 | 0.1011 | 0.1070 | 0.1130 | 0.1188 |
| 0.1 | 0.0980 | 0.0798 | 0.0768 | 0.0819 | 0.0880 | 0.0941 | 0.0997 | 0.1058 | 0.1119 | 0.1178 |
| 0.25 | 0.0709 | 0.0598 | 0.0658 | 0.0732 | 0.0808 | 0.0879 | 0.09433 | 0.1009 | 0.1075 | 0.1139 |
| 0.5 | 0.0445 | 0.0555 | 0.0629 | 0.07098 | 0.0789 | 0.0863 | 0.0929 | 0.0997 | 0.1064 | 0.1128 |
| 1 | 0.0504 | 0.0619 | 0.0677 | 0.0748 | 0.0821 | 0.0890 | 0.0953 | 0.1018 | 0.1083 | 0.1146 |
| 1.5 | 0.0696 | 0.07479 | 0.0773 | 0.0825 | 0.0885 | 0.09454 | 0.1001 | 0.1061 | 0.1122 | 0.1181 |
| 2 | 0.0791 | 0.08115 | 0.0821 | 0.0863 | 0.0917 | 0.0972 | 0.1025 | 0.1082 | 0.1141 | 0.1198 |
| 2.5 | 0.0825 | 0.0834 | 0.0838 | 0.08769 | 0.0928 | 0.09824 | 0.1033 | 0.1090 | 0.1148 | 0.1204 |
| 5 | 0.0851 | 0.0851 | 0.0851 | 0.0887 | 0.0937 | 0.0989 | 0.1039 | 0.1095 | 0.1153 | 0.1209 |

Table 1: Average tests errors for several choices of $p$ and $\beta$ used for parameter tuning. No clustering has been performed. The average test error we obtain for MNL is 0.168.

| | $\beta = 0.05$ | 0.1 | 0.25 | 0.5 | 0.75 | 1 | 1.25 | MNL |
|---|---|---|---|---|---|---|---|---|
| $p = 0.1$ | 0.0997 | 0.0771 | 0.0732 | 0.0748 | 0.0785 | 0.083 | 0.0876 | 0.1877 |
| $p = 1.25$ | 0.0788 | 0.09273 | 0.1022 | 0.1116 | 0.1224 | 0.1324 | 0.1409 | 0.1883 |

Table 2: average test error of learning choice probabilities, comparison of Mallows model with different $\beta$s and MNL; 2 clusters are generated based on distance function $\mathcal{K}^p$ for given $p$. The best values are highlighted.

## B.2 Runtime of PRIM and DYPCHIP

| | pre-processing time | | | | | amortized time per sample | | | | |
|---|---|---|---|---|---|---|---|---|---|---|
| | $k = 8$ | $k = 10$ | $k = 12$ | $k = 14$ | $k = 16$ | $k = 8$ | $k = 10$ | $k = 12$ | $k = 14$ | $k = 16$ |
| $n = 200$ | 0.007 | 0.035 | 0.19 | 1.06 | 8.64 | 5.67e-05 | 9.59e-05 | 0.00022 | 0.00074 | 0.0032 |
| $n = 500$ | 0.008 | 0.044 | 0.23 | 1.24 | 7.83 | 7.89e-05 | 0.00012 | 0.00025 | 0.00076 | 0.0028 |
| $n = 1000$ | 0.012 | 0.079 | 0.29 | 1.46 | 8.39 | 0.00011 | 0.00022 | 0.00031 | 0.00082 | 0.0030 |

Table 3: average (among 10 runs) runtime of sampling algorithm PRIM in seconds, $\beta = 0.2$

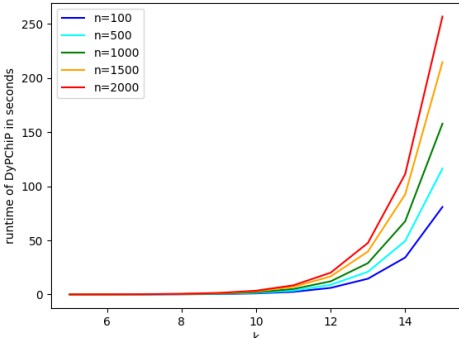

Figure 3: Runtime of DYPCHIP (in seconds) for various choices of $n, k$. Here $\beta = 0.6$, $p = 0.5$ and $w = 2\vec{1}$, size of assortment $r = k - 2$.

## B.3 Additional tables and figures for the analysis of sample complexity of BUCCHOI and FINDTOP on synthetic data

| | $k = 6$ | $k = 10$ |
|---|---|---|
| $n = 1000$ | $6.95 \pm 5.06$ | $0.4 \pm 0.48$ |
| $n = 5000$ | $228.5 \pm 53.3$ | $228.6 \pm 64.56$ |
| $n = 10000$ | $934.05 \pm 209.35$ | $870.1 \pm 259.64$ |
| $n = 20000$ | $3505.05 \pm 409.87$ | $3282.55 \pm 708.9$ |

| | $k = 6$ | $k = 10$ |
|---|---|---|
| $n = 1000$ | $0.1 \pm 0.3$ | $0.0 \pm 0.0$ |
| $n = 5000$ | $0.35 \pm 0.50$ | $1.92 \pm 0.92$ |
| $n = 10000$ | $2.95 \pm 2.28$ | $3.0 \pm 3.16$ |
| $n = 20000$ | $5.1 \pm 3.5$ | $9.0 \pm 13.2$ |

Table 4: Kendall's Tau distance of true and learned center by BUCCHOI with 100 (left table) or 200 (right table) samples for $n = 1000, 5000, 10000, 20000$ and $k = 6, 10$. We let $\beta = 0.6$. distance reported as $mean \pm std$ these numbers are taken from 10 runs of the algorithm.

| | $\beta = 0.4$ | $\beta = 0.6$ | $\beta = 0.8$ | $\beta = 1$ | $\beta = 1.2$ |
|---|---|---|---|---|---|
| 50 samples | $12.75 \pm 4.4$ | $8.25 \pm 7.7$ | $7.05 \pm 5.64$ | $4.35 \pm 7.5$ | $0.7 \pm 1.65$ |
| 100 samples | $3.5 \pm 0.67$ | $0.35 \pm 0.549$ | $0.0 \pm 0.0$ | $0.0 \pm 0.0$ | $0.0 \pm 0.0$ |

Table 5: Kendall's Tau distance of true and learned center by BUCCHOI with 50 and 100 samples for various $\beta$, $n = 1000$, $k = 12$, $p = 0.5$ reported as $mean \pm std$. mean and standard deviation is taken for 10 runs of the algorithm for each set of parameters.

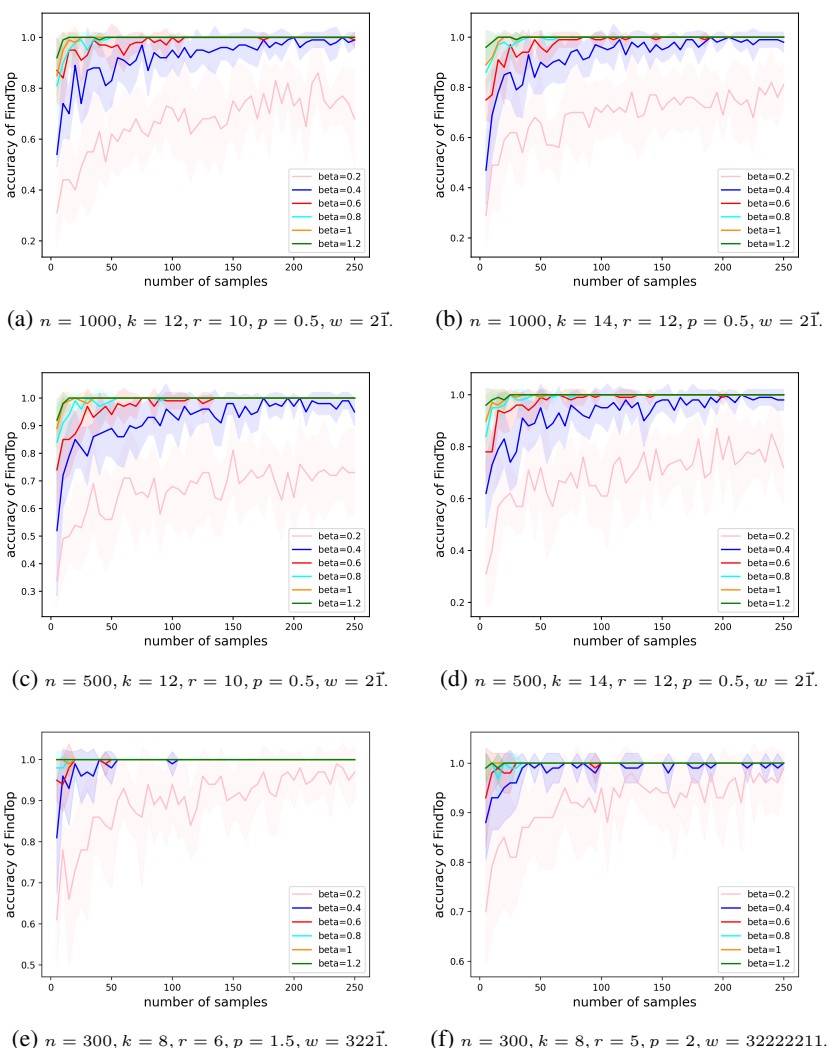

(a) $n = 1000, k = 12, r = 10, p = 0.5, w = 2\vec{1}$.

(b) $n = 1000, k = 14, r = 12, p = 0.5, w = 2\vec{1}$.

(c) $n = 500, k = 12, r = 10, p = 0.5, w = 2\vec{1}$.

(d) $n = 500, k = 14, r = 12, p = 0.5, w = 2\vec{1}$.

(e) $n = 300, k = 8, r = 6, p = 1.5, w = 322\vec{1}$.

(f) $n = 300, k = 8, r = 5, p = 2, w = 32222211$.

Figure 4: Sample complexity of FINDTOP for a wide range of parameters.

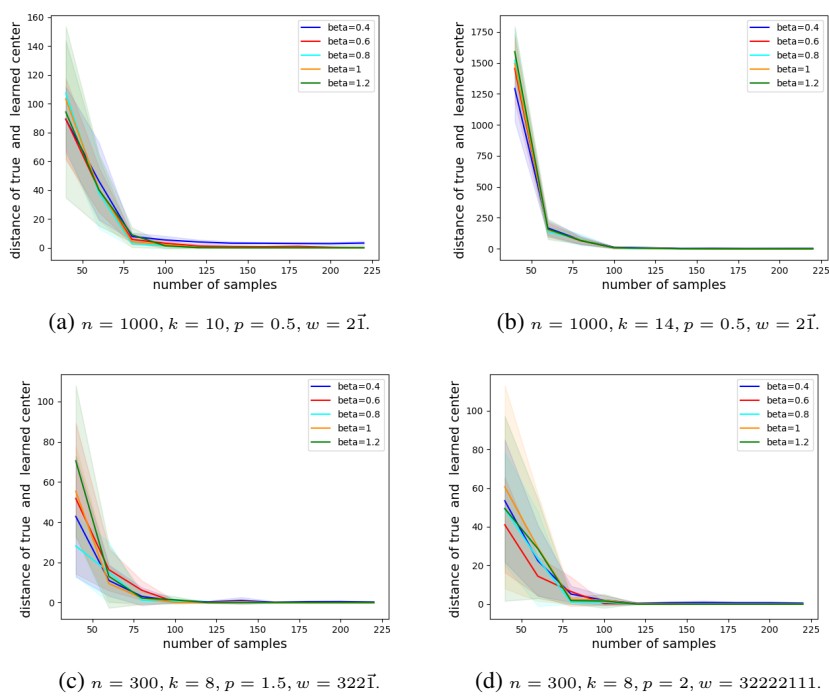

(a) $n = 1000, k = 10, p = 0.5, w = 2\vec{1}$.

(b) $n = 1000, k = 14, p = 0.5, w = 2\vec{1}$.

(c) $n = 300, k = 8, p = 1.5, w = 322\vec{1}$.

(d) $n = 300, k = 8, p = 2, w = 32222111$.

Figure 5: Sample complexity of BUCCHOI for a wide range of parameters. y axis shows the Kendall's Tau distance between true and learned center while x axis shows sample complexity

