# OpenReview forum: "Generalized Top-k Mallows Model for Ranked Choices"
_NeurIPS.cc/2025/Conference — NeurIPS 2025 spotlight_

### Official Review · Reviewer_jMHU · 2025-06-11

**Clarity:** 2
**Significance:** 3
**Originality:** 3
**Rating:** 5
**Confidence:** 3

**Summary:**

This paper proposes inference methods for generalized Mallows model using top-K ranking data, in which each item has an associated weight. Inference is built on efficient sampling of rankings by reducing the computation time of the method by Chierichetti et al. (2018a). Two methods are then proposed for estimating choice probabilities given model parameters and estimating model parameters given choice data.

**Questions:**

N/A

**Ethical Concerns:**

["NO or VERY MINOR ethics concerns only"]

**Final Justification:**

The authors have addressed all of my comments

**Limitations:**

Please see above.

**Quality:**

2

**Strengths And Weaknesses:**

Strengths: Extensive theoretical analysis and synthetic data experiments are provided. Real dataset experiment shows improvement against the Multinomial Logit (MNL) model.

Weaknesses:

A notation table would have been very helpful to improve clarity, particularly as the proofs in the Appendix have some inconsistencies (e.g., caligraphic P should be standard P in line 815, Equation 6 numerator should be A^bar). Further, terminology is inconsistent such as items, samples, products meaning the same thing.

While it is important to outperform MNL, this observation is somewhat expected with respect to the literature (e.g. Desir et al.). It is more interesting to compare the proposed algorithms in terms of predictive accuracy and computation time to existing efficient top-k Mallows model inference algorithms like Chierichetti et al. or Fotakis et al.

Hyperparameters p and beta were tuned on the test set (rather than a separate validation set), which is unfair against the MNL model.

---

> ### Author Rebuttal · Authors · 2025-07-30
>
> Thank you for your comment on the clarity of our paper. We will fix the typos and include a notation table in the camera-ready version. We would appreciate engagement in the rebuttal so we can address the reviewer's concerns.
>
> We now respond to raised questions:
>
>
> **Comparison with Chierichetti et al. or Fotakis et al.**
>
> A direct experimental comparison with Chierichetti et al. and Fotakis et al. is not straightforward because our work addresses a fundamentally different problem. Our paper focuses on the choice modeling setting, using our BUCCHOI and DYPCHIP algorithms to learn a preference model from choice data—i.e., observing a single selected item from an assortment. Chierichetti et al. learn the central ranking from a dataset of observed top-k lists, which is a different and more information-rich signal than choice data. Fotakis et al. focus on the problem of aggregating incomplete and noisy rankings to find a consensus ranking; a distinct task from learning a generative model to predict single choices from assortments. Comparing predictive accuracy would require extending these models to handle choice data, a non-trivial task that our work addresses. To make this distinction clearer, we will expand our Related Work section to better contextualize our contributions.
>
>
> **Hyperparameters $p$ and beta were tuned on the test set (rather than a separate validation set), which is unfair against the MNL model**
>
> Our goal was to do a comprehensive study based on different choices of $p$ and $\beta$. To this end, we performed experiments for these different choices (through a grid search). Our goal was not to do tuning rather a comprehensive study.  We can clarify this better in the final version. We would like to emphasize that even without tuning we get better test errors; the test error of our algorithm is at most 0.11 along all parameter choices while MNL test error is 0.168. Please see table 1 in the appendix.

---

> > ### Comment · Reviewer_jMHU · 2025-08-06
> >
> > The authors effectively responded to all of my comments, my updated rating is Accept, thank you

---

### Official Review · Reviewer_7DBg · 2025-06-26

**Clarity:** 2
**Significance:** 3
**Originality:** 2
**Rating:** 5
**Confidence:** 2

**Summary:**

The paper studies top-K Mallows model, a variant of the ground truth-based statistical model, for sampling rankings. The paper builds on a paper by Chierichetti et al. titled “Mallows models for top-k lists”. First, the paper provides the “PROFILE BASED REPEATED INSERTION METHOD”, i.e.,  sampling procedure for the generalized version of the Mallows model. Second, the paper introduces the “DYNAMIC PROGRAMMING FOR CHOICE PROBABILITIES”, i.e., a method for calculating the choice probabilities functions. Third, the paper provides the “BUILD CENTER FROM CHIOCES”, i.e., a learning algorithm for inferring the ground truth of Mallows. Finally, the authors provide experimental analysis, demonstrating the behavior of the top-k Mallows model. They use synthetic data and real-life Sushi model (one of the most frequently used dataset in computational social choice).

**Questions:**

Q1: In what practical use-case the p parameter would be set to a value larger than 1? (Intuitively I would say that p should fall in [0,1] interval).

**Ethical Concerns:**

["NO or VERY MINOR ethics concerns only"]

**Final Justification:**

Considering the entire discussion, I increase the evaluation from "borderline accept" to "accept".

**Limitations:**

Yes.

**Paper Formatting Concerns:**

The font size in Figures 1 and 2 is far too small.

**Quality:**

3

**Strengths And Weaknesses:**

[Significance]

The Mallows model is widely used, and the generalization to the top-*k* variant seems like a natural and valuable extension that one would want to have.

[Originality]

The originality of this paper is somewhat limited, although I don't see that as a major drawback.

[Quality]

The overall quality of the paper is good, with some minor issues that should be addressed:

- Lines 138/139 “This concept has been extended to top-k lists, where it no longer forms a true metric but retains useful mathematical properties.” This statement is too vague. There are various ways in which a function can fail to be a true metric—for example, becoming a pseudometric, semimetric, or quasimetric. It seems that the function described here is a semi-metric as it fails the triangle inequality. Please clarify explicitly in what way the function deviates from being a true distance.
- The citations occasionally feel somewhat arbitrary. While the paper includes an extensive list of works related to the Mallows model, several cited papers seem only vaguely relevant. It would be helpful if the authors reviewed the bibliography more carefully to ensure relevance and accuracy. For example, ”Liu, A. and Moitra,” appears twice. (Side comment: it might be worth taking a look at “Properties of the Mallows Model Depending on the Number of Alternatives: A Warning for an Experimentalist”)
- “the generalized Mallows model for top-k lists provides a more realistic framework
for understanding user preferences”—while this is quite a strong statement, there is little evidence supporting it in the paper.
- The authors state that “the Multinomial Logit (MNL) (Bradley & Terry, 1952) model being the most widely used due to its simplicity and interpretability.” Please provide justification for that claim. Additionally, clarify the context in which the model is considered “most widely used”. For example, in the context of computational social choice, the “Guide to Numerical Experiments on Elections in Computational Social Choice” suggests that the impartial culture model is the most widely used.

[Clarity]

Clarity of the paper could be improved.

- The font size in Figures 1 and 2 is far too small. As a result, the plots are difficult to read and fall short of NeurIPS presentation standards. Please increase the font size and improve the overall readability of the figures.
- The “Preliminaries and Definitions” section looks like it is almost copy pasted from the “Mallows models for top-k lists” paper.
- The authors refer to Kendall’s tau distance, by writing: “Kendall”, “Kendall tau, and “Kendall Tau”. Please unify the name throughout the paper.
- Line 26: “The Mallows model Mallows (1957)” is awkward. Please rephrase this sentence or use a different citation command.
- Line 167: A misplaced comma should be moved one line above
- Line 179: “method for permutations Doignon et al” is awkward. Please rephrase this sentence or use a different citation command.
- Typo in line 115: CHIOCES

---

> ### Author Rebuttal · Authors · 2025-07-30
>
> Thank you for your time and review. We make sure to apply the suggested changes to make the paper more clear. Please see below for responses to your questions.
>
> **Question.  In what practical use-case the p parameter would be set to a value larger than 1?**
>
> $p \\geq 0$ is the parameter for controlling the penalty per a pair of elements that are comparable in one top-k ranking but incomparable in the other.
>
> Consider $\tau$ to be the top-k center. For instance, let $n=100$ and $k=10$, and consider the center $\tau=1, 2, 3, 4, 5,...10$. By increasing $p$ (e.g., $p>1$), the probability of any top-k ordering $\sigma$ with a small size of $\tau \cap \sigma$ will decrease. For instance, in $\sigma=11,12,13,14,...,20$ or $\sigma=12,13,...21$, or any other top-k with $\tau \cap \sigma=\emptyset$, there are ${10}\choose{2}$ elements that are comparable in $\tau$ and incomparable in $\sigma$, all of which incur the penalty of $p$. Meanwhile, in $\sigma=10, 9, 8,...1$, all pairs pay a penalty of $1$. Therefore, by increasing $p$, we decrease the probability of top-k's whose intersection with the top-k center is small.
>
> The use case where $p > 1$ would correspond to cases where losing information regarding a relative order of two items is much worse than mis-ranking them, i.e., inclusion of an item in the top-k set is more critical than its exact position within that set.
>
> A practical example is a product recommender system on an e-commerce website. Imagine a system that displays the "top 5 laptops for students." For a potential customer, it is crucial that the most relevant and suitable laptops appear on this initial list. If a highly appropriate laptop fails to make the top-5 cut entirely, the user may never discover it, leading to a poor user experience and a potentially lost sale. This large penalty for an item dropping out of the top-k list corresponds to a high value of $p$.
>
> In contrast, if that same laptop is present in the top-5 list but is ranked at position #4 instead of its "true" rank of #2, the outcome is still largely positive. The user sees the item, can click to evaluate its specifications, and might still purchase it. The penalty for this internal mis-ranking (an inversion) is less severe.
>
> **Comment on the importance of the impartial culture model versus MNL:**
>
> In the  impartial culture IC, the preference of each user is drawn independently and uniformly from the rankings of candidates. The Impartial Anonymous Culture (IAC) complements this model by using a uniform distribution over preference profiles (defined as tuples of rankings).
> Both of these models are primarily used in analysis of voting behaviors. In many applications, the simplicity of these models (in particular the assumption of uniform samples) is unrealistic therefore, the MNL model is used in a wider range of applications.
> We note that the IC model (uniform sampling) is a special case of the MNL and Mallows model. In MNL we obtain uniform distribution by setting all utility values equal to one and in Mallows model we obtain it by setting the discrepancy parameter $\beta$ equal to 0
>
>
> **The section 2 looks like it is almost copy pasted from the “Mallows models for top-k lists” paper:**
> There are some similarities, although we state these definitions in a more general form and use inversions tables. In this section, we review the definition of Mallows models for top-k lists. Note that the main contribution of our paper is to use this model for modeling choice and choice probabilities.

---

> > ### Comment · Reviewer_7DBg · 2025-08-04
> >
> > I thank the authors for addressing my comments.

---

### Official Review · Reviewer_wgQk · 2025-07-02

**Clarity:** 3
**Significance:** 2
**Originality:** 3
**Rating:** 5
**Confidence:** 4

**Summary:**

The paper proposes a generalized Mallows model for top-k preference lists over n items that allows item-specific weights and extends the Kendall-tau distance to partial rankings. Three tasks are considered: sampling a top-k list according to the model. It takes  O(k*2^k+klogn) time; computing the choice probabilities, which takes O(2^k*k^3*∣A∣^2) time, where A is the assortment; and an active learning algorithm that learns the central ranking with O(log n) assortments.

**Questions:**

1 It is claimed that "once setting w = 1, we obtain Equation (TopKMM)." But that equation is actually parameterized by w. Is there a typo?

2. Why omit Mixed-MNL or Plackett–Luce mixtures? This is usually included in the literature so explanation of the exclusion would be helpful.

3. It is unclear what the 100-sushi data tries to mimic in practice. If the authors are just trying to mimic a situation where the observer can only see single choices, then the value of "k" is only a hyper-parameter to be tuned. If the authors are actually using the fact the the raw data is based on top-10 lists, why not directly train the model based on the top-10 lists?


Apart from that, I do not have questions beyond the "weakness" comments.  That said, in the rebuttal, if the authors can come up compelling advantages of the newly proposed model/method compared to what has been proposed in the literature, I am happy to re-evaluate.

**Ethical Concerns:**

["NO or VERY MINOR ethics concerns only"]

**Final Justification:**

Overall, the Mallows model and its variations remain challenging combinatorial objects to analyze and apply. This paper makes a meaningful contribution by extending the literature to a weighted version of top-k distributions. The authors’ analysis of choice probabilities from arbitrary assortments, along with their demonstration of how the model can be applied to learning from discrete choice data, is particularly commendable.

In my view, the contributions are sufficient. I am happy to recommend acceptance.

**Limitations:**

Yes (after various clarifications of the rebuttal process).

**Paper Formatting Concerns:**

NIL

**Quality:**

3

**Strengths And Weaknesses:**

Strength:


The main contribution of this paper is the introduction of a probabilistic model for top-k lists, which extends the Generalized Mallows Model of Fligner and Verducci (1986). The analysis of the sampling procedure and the derivation of choice probabilities build naturally on prior work, particularly the RIM method and the dynamic programming framework. The derivations can be difficult, given the complex combinatorial structure, and hence the paper has merits in terms of technical contributions. Overall, the paper is clearly written.


Weakness:



1. (Computational complexity) It is unclear how computationally tractable the proposed GMM for top-k lists is in practice. For example, according to Theorem 3.1, even sampling a top-k list can take O(k·2^k) time, which is a very weak guarantee when k is close to n. This contrasts with the RIM method, which remains efficient at least when k = n, and the sampling procedure of Feng and Tang (2021), which is very efficient albeit based on a different variant of the Mallows model. Unsurprisingly, computing choice probabilities also appears challenging when k is large; see Theorem 4.1.


2. (Parameter learning from choice data) Given the discussion above, it is only natural to expect that learning (i.e., estimating the model parameters) from choice data would be even more difficult. I therefore read Section 5 with interest, but found the results somewhat underwhelming for several reasons:


    - 2a First, the analysis assumes that the algorithm has full control over the assortments. In fact, I believe BUCCHOI only picks assortments of size at most two. From a practical perspective, however, the observer may only be endowed with observed data and then the question is how to infer the model parameters (e.g., the central ranking) in which the assortment history is arbitrary (i.e., not controlled by the observer). Methodologically, this is precisely the challenge that much of the prior literature (e.g., Desir et al. 2022; Feng and Tang, 2021) was trying to address, but this paper effectively sidesteps it. If the assortments are only pairwise, there is already a **very** vast body of work on passive and/or active learning of ranking and/or best-item identification from pairwise comparisons. (So even in the active learning setting, it is unclear how strong the sample complexity guarantee of \Theta(\log n) is.)


    - 2b Another important question is that the model brings richness by introducing a few new parameters and the paper remains relatively silent about how to determine them. Based on Section 5, the underlying data are choice data (rather than ranked or top-k list data). This makes the value of k essentially a modeling parameter, but it is unclear how to "learn" the right k. In addition, one of the key distinctions of this model compared to the original Mallows model is the introduction of the "weights" w. However, it remains silent how these w parameters should be calibrated. If making them uniformly one, does the proposed model go back to some more basic version of TopKGMM?

---

> ### Author Rebuttal · Authors · 2025-07-30
>
> We thank the reviewer for their helpful comments; we will use this feedback in preparing the camera ready version.
> Here is our response to some of the questions:
>
> **Re W1 (Computational complexity)**
>
> The sample complexity of $O(k·2^k)$ is only for a pre-processing step in PRIM (sampling algorithm), and generating samples after this step only takes polynomial time. We assume that $k$ is small, which is a common assumption due to practical setups in online platforms, and cognitive abilities of users for ranking items (see Chierchetti et al for more explanation). A common choice for $k$ is $10$ (with datasets collected as in thetoptens website)  for which the preprocessing time is constant and hence generating samples takes $\Theta(\log n)$ (see Theorem 3.1). We also note that our algorithm is more efficient than the sampling algorithm of Chierichetti et al (even considering the pre-processing step).
>
> Both in the preprocessing step of RIM and DypCHIP the $2^k$ term is an upper bound on the number of ``profiles’’. For $k$ close to $n$ the number of profiles becomes polynomial in $n$ and specifically for $k=n$ the number of profiles is only 1. For $k=n$ PRIM and RIM are the same algorithm since we will only have one profile.
>
> **Re W2 (Parameter learning from choice data)**
>
> **Re W2a**
>
> We would like to clarify a few points about our learning algorithm BUCCHOI: (1) whether it is only applicable in active learning scenarios (2) how does the algorithm compare with prior works which use pairwise comparison data.
>
> The crux of our active learning algorithm BuCCHOI is the FindTOP (Algorithm 4 in the appendix). Given an assortment $A$  which is repeatedly shown to the user and choice data collected,  FindTOP finds: an element in the assortment which has the highest rank among other elements in the assortment w.r.t. $\tau^\*$ the center of the distribution. If such an element does not exist, i.e., none of the elements in the assortment are in the center (thus, they are all incomparable w.r.t. $\tau^*$ ) the algorithm returns none.
> A simple case is if the assortment  $A=\\{i,j\\}$ has size two. In this case FindTOP outputs  $i$ meaning $i>\_{\tau^\*} j$ or outputs $j$ meaning  $j>\_{\tau^\*} i$ or None meaning  $i\parallel_{\tau^\*} j$. Thus, we obtain full information about the relative order of $i$ and $j$.
>
> If the assortment $A$ has size $\\ell$ we either infer $\\ell-1$ pairwise comparisons of the elements; or that all $\ell$ elements are incomparable and do not belong to the top-k center. For instance, assume given $A$ FindTOP returns $i\in A$  this means that $i>_{\tau^\*} j$ for any $j\\in A\setminus\\{i\\}$ ($\\ell-1$ relations) or it returns None meaning $i,j\in A\implies i\\parallel\_{\\tau^\*} j$.
>
> FindTOP can be used either in a passive algorithm where the assortments are not selected by the algorithm or in an active algorithm. Clearly, if the assortments are not selected, we may only obtain some partial knowledge of the top-k center. It is not clear how to then expand this partial information to a top-ranking. We will add this question as future work in our conclusion and we will clarify that our methodology is an active algorithm in the introduction.
>
> The reviewer is correct that  BuCCHOI is only using pairwise comparison. But this is only for the simplicity of design. We could use FindTOP with various sets of assortments to find the center. For instance assume that the assortments must be all of size $\ell$. In this case we can use the following algorithm:
>
> * Divide the n elements to non-intersecting assortments of size $\\ell$ (assuming that n is divisible by ell for simplicity):
> * Run FindTop on these assortments, let $S = \\emptyset$
>  * * If FindTOP returns NONE then none of the elements in the assortment are in the center. Add them to set $S$.
>  * * If FindTOP returns $i$, $i$ belongs to the center, replace $i$ with an element $j$ in $S$ and run FindTOP on this new assortment.
> * Repeat the above two steps until all elements of [n] are either in $S$ and for those which are not in $S$ all the pairwise orders are determined.
>
> Note that this procedure concludes after $n/\\ell+2k$ assortment queries as with $n/\\ell+k$ calls we can get the top-k elements and with $k$ calls we can get the ordering of them.
> Similar to Theorem 5.1 we can prove that the sample complexity will be $O(\\ell^2\\log n)$.
>
> The reason that we design BUCCHOi using pairwise comparisons is merely simplicity and many other algorithms similar to the above can be designed to infer the relative orders of all pairs of items.
>
> We thank the reviewer for mentioning the existence of a broad body of works considering pairwise comparisons. When it comes to learning Mallows model center the most notable prior works are [Lu and Boutilier] and [Vitelli et al]. Both of these works are based on Maximum likelihood estimation and the EM method and only the convergence of algorithms are shown. These works differ from our work in the sense that they do not include any mathematical analysis of finite sample complexity bounds or theoretical guarantees on the error.
>
> In our work by the use of concentration bounds we have provided a mathematical guarantee on the error of FindTOP and BuCChoi showing their finite sample complexity,
> This kind of mathematical guarantee does not exist in prior work which considers choice data or pairwise data and only exists when permutations or top-k samples are available or a linear order of some items are available.
> We thank the reviewer for pointing out this ambiguity and will add the above discussion to our paper.
>  - [Lu and Boutilier] Effective Sampling and Learning for Mallows Models with Pairwise-Preference Data, Tyler Lu and Craig Boutilier, JMLR 2014
> -  [Vitelli et al] Probabilistic Preference Learning with the Mallows Rank Model, Valeria Vitelli
> Oystein Sorensen, Marta Crispino, Elja Arjas, JMLR 2018
>
>
> **Re W2b**
>
> BuCCHOI is learning both the center and $k$. BuCCHOI (or any other algorithm which uses FindTOP such as the above) finds all pairwise ordering of the elements in [n] without knowing $k$. When these orderings are found, k is also found. Therefore, BuCCHOI also learns k. The reviewer is correct that we do not have an algorithm for learning the weights however we are **not** assuming that $\\vec{w}=1$. In order for our algorithm to work (Theorem 5.1), we just need that $\\beta> \\log 3/w\_{min}$. Learning weights requires additional analysis and hence we leave it as a future direction.
>
>
> **Re Q1**
>
> We are not sure what you are referring to here. We don’t have $w$ in line 143. Would you please clarify where you are referring to?
>
> **Re Q2**
>
> We considered the possibility of mixture models however, our algorithm is not directly applicable to mixture models. To handle a mixture of models, we first cluster the data to 2-5 clusters. Our thought for bypassing MNL mixture models was that we wanted to have an apple to apple comparison with our method which is not a mixture model.
> We note that, after doing the clustering the silhouette coefficient was always very low and we observed the lowest test error of our algorithm associated with no clustering. Therefore, we did not explore other possibilities for mixture models.
>
>
> **Re Q3**
>
>  It is true that the Sushi dataset is top-10 and we can just focus on learning top-$k$ without assuming access to choice data, however the goal of our paper is to use choice models to learn top-$k$. Since this is a prominent dataset used in prior work and that we were not aware of any other dataset suitable for our method, we curated it to generate choices.
> We assume that the top-10 lists of the sushi data set are samples generated from a TopKMM and we learn its center. We have used these samples and our definition for choice for top-k lists (lines  231-233 of the submitted manuscript) to generate choice data. In particular, for a given (we chose $A$ randomly) assortment $A$ and sample $\tau$ we take the choice to be the highest element of $A$ with respect to $\tau$ if $A\cap \tau=\\emptyset$ we select uniformly at random an element of $A$.

---

> > ### Author Response · Authors · 2025-08-04
> >
> > Dear Reviewer wgQk,
> >
> > We would like to know if our rebuttal has adequately addressed your concerns. Please let us know if you have any remaining questions or would like to discuss any points further through an interactive discussion.

---

> ### Comment · Reviewer_wgQk · 2025-08-04
>
> I thank the authors for their detailed reply. I have a few follow-up comments and clarifications.
>
> Re W1 (Sampling Tractability):
> My understanding is that sampling from the TopKGMM model is significantly less tractable than from the Mallows model. For instance, it is well known that one can efficiently sample full rankings from the Mallows distribution, but Theorem 3.1 suggests that it is not the case under TopKGMM. The “Profile-based” TopKGMM distribution appears to be a different (and arguably simplified) variant of the original problem. If the authors disagree with this assessment, a natural rebuttal would be to provide/point to an efficient sampling algorithm for the general TopKGMM.
>
> I would like to note that this is not necessarily a fatal flaw, but potentially noteworthy since sampling is arguably one of the easiest tasks when it comes to ranking models. Various ranking models allows for efficient sampling at the very least; see references above. Given that even sampling appears intractable here, it becomes perhaps less surprising that other tasks, such as computing choice probabilities, may also require time exponential in k—in contrast to, say, the Mallows model.
>
>
>
> Re W2 (Learning from general choice data)
>
> Regarding learning from passive choice data, the authors may consider clarifying the following central question more directly: Given (arbitrary) choice data {(c_1, A_1), (c_2, A_2), ..., (c_T, A_T)}, how to infer the model parameters (e.g., central ranking and k) from the choice data? The authors seem to claim that they can do it (both in the paper and in the rebuttal). But what I can find is various simplified versions of this task. For example, FINDTOP focuses on a fixed assortment. BUCCHOI (i) assumes that the learner can choose assortments and (ii) focuses on pairwise comparison. Please feel free to clarify if I have overlooked any general inference result.
>
>
> (On Related Work:)
>
> The authors state that there is little work on error quantification for Mallows models and their variants. I would like to point to a few starters. For example, [Tang, Wenpin. "Mallows ranking models: maximum likelihood estimate and regeneration." International Conference on Machine Learning. PMLR, 2019] studies the statistical properties of the MLE for the Mallows model, and [Feng, Yifan, and Yuxuan Tang. "A Mallows-type Model for Preference Learning from (Ranked) Choices." Available at SSRN 4539900 (2023)] explores both statistical and sample complexity properties of a (arguably simpler) variant of Mallows model.
>
> In the meantime, I wish to point out that even for pairwise comparison under the Mallows model, sample complexity guarantees could perhaps be easier to derive if the learner has the freedom to choose the assortments.
>
>
>
>
> Q1: Resolved—thank you.
> Q2: This seems to be a schematic issue: any RUM is, in a technical sense, a mixture over rankings. In that regard, even MNL can be viewed as a mixture model (of ranking distributions).
> Q3: Two questions:
> (i) If the authors are generating single choice data from the SUSHI dataset, why not use the 10-SUSHI version with full rankings, which is more commonly used in the literature?
> (ii) If the parameter k is to be learned, what value is recovered from the SUSHI data?

---

> > ### Author Response · Authors · 2025-08-05
> > **Re: rebuttal response**
> >
> > Thank you for your thoughtful feedback and the opportunity to clarify our work. We appreciate your insightful comments.
> >
> > **Re Re W1**
> >
> >  We would like to clarify that the Profile-based Repeated Insertion Method (PRIM) is an exact sampling algorithm for the general TopKGMM, not a simplified variant. Our method partitions the sample space by profiles ($S$), but as demonstrated in the proof of Theorem 3.1, this technique correctly recovers the target TopKGMM distribution for any top-$k$ list $\tau$ . This approach was developed to solve the open problem of creating an efficient, RIM-like sampler for top-$k$ lists, a challenge highlighted by Chierichetti et al. (2018a) . We will revise the manuscript to state more explicitly that our method samples from the original, unmodified distribution
> >
> >
> > The algorithm's O(k2^k) complexity, while exponential in $k$, is a significant improvement on the previous O(k^2 4^ k) bound from prior work and remains practical for the small k values common in real-world applications . This inherent complexity is a necessary trade-off for the model's greater expressiveness compared to the classic Mallows model. As you noted, this complexity extends to other tasks like computing choice probabilities, which is expected for this more challenging top-k setting. Our contribution is in providing the first algorithms capable of tackling these computationally intensive tasks.
> >
> > In case your comment also asks about our runtime for sampling full rankings, we would like to add that for larger $k$, such as $k=n-\ell$, a rough runtime upper bound is about $\ell*n^\ell$. This is because only profiles of size at least $n-2\ell$ can be extended to a feasible top-$k$ as there are at most $\ell$ items that are not in the central top-$k$ and can be added to the profile; so the number of different profiles are ${n-\ell \choose 1} + {n-\ell \choose 2} + ….+{n-\ell\choose \ell} \leq \ell (n-\ell)^\ell$. Particularly for $k=n$, our runtime is $O(n)$ which matches the runtime of RIM for classic Mallows model.
> >
> >
> > **Re Re W2**
> > We suggest the following pseudocode for passive algorithms which receives as input data as suggested by the reviewer and outputs a DAG corresponding to the center:
> >
> >
> > Input: ${\mathcal D}= {(c_1, A_1), (c_2, A_2), ..., (c_T, A_T)}$
> > - Organize $ {\mathcal D}$ into buckets $B_1, B_2,\dots B_t$ so that in each bucket the given assortment is fixed.
> > - $G= n\times n$ matrix showing the adjacency matrix of a DAG corresponding to the recovered center
> > - For $i=1 :t$
> >  - - $A$= assortment of $B_i$
> >  - - $C_i$= sequence of choices in $B_i$
> >  - - c=FindTOP(C_i, A)
> >  - - For $a\in A$:
> >  - - - If $c\neq a$:
> >  - - - - $D_{ca}=1$
> >  - - - - $D_{ac}=0$
> > - Return G
> >
> > If each assortment is presented enough number of times (see lemma 5.2) and DAG has enough edges to recover the center then the center can be constructed.
> >
> > **Re Related work**
> > Thanks for pointing out that Tang ICML 2019 includes error quantification results.  We will add this to the related work of our paper. The work of Feng and Yuxuan NeurIPS 2022 considers a Mallows like distribution, which uses a distance measure which is **not** defined in terms of the number of inversions; thus it is inherently different from Kendall Tau distance used in Mallows model which has a more complicated structure.
> >
> > The reviewer is correct that for pairwise comparisons, reconstruction of the center and deriving the  complexity guarantees are easier. We would like to add that we can employ FindTOP to a wider range of assortments and derive similar sample complexity results. For instance, the pseudocode we have included  in the rebuttal actively shows to the users assortments of size $\ell$ ($\ell$ can be chosen arbitrarily) and similar to Theorem 5.1 we can show  that its sample complexity is $\ell^2 \log n$.
> >
> > **Re Q3 additional questions**
> >
> > (i) The top-10 list version of the SUSHI dataset was intentionally chosen because our paper's primary contribution is the TopKGMM, a model specifically designed for top-k lists rather than full rankings. Using a dataset that naturally exists in a top-k format provides the most direct and appropriate evaluation of our proposed model's performance and utility in a real-world scenario. While the full ranking version could be truncated to create top-k lists, using the existing top-10 data aligns perfectly with the problem domain our work addresses.
> >
> > Another point that we would like to note is that the choice $c$ from a top-k list are defined as follows:
> > - If $A\cap \\tau^\*\neq \\emptyset$, let $c$ be the the highest element of $A$ w.r.t $\tau^\*$.
> > - If $A\cap \\tau^\*= \\emptyset$, let $c$ be an element of $A$ picked uniformly at random.
> >
> > Note that if we have used the 10-sushi version case 2 would never happen so we always pick (deterministically) the top element of $A$ w.r.t. $\tau^*$.
> >
> > (ii) In the center we have learned $k=14$.

---

> > > ### Comment · Reviewer_wgQk · 2025-08-05
> > >
> > > I thank the authors for their further clarifications. My earlier questions regarding (i) the tractability of sampling full rankings from the proposed model and (ii) parameter estimation from observed choice data have been satisfactorily addressed. Congratulations on the nice work.
> > >
> > > I have two other questions/concerns that remain unaddressed.
> > >
> > > 1. In light of the new discussion, it is still unclear how to compute choice probabilities from arbitrary assortments when k = n. Note that under the Mallows model, this is tractable. My understanding is that under the new model, it is not. The authors are welcome to rebut this by directly explaining the complexity of the choice probability calculation under the new model when k = n.
> > >
> > > 2. Across the paper and rebuttal, there appears to be some conflation between distinct meanings of the term "top-k":
> > >
> > > - The first "top-k" is a model parameter, i.e., a probability distribution over top-k lists, which is the focus of this paper’s theoretical contributions.
> > >
> > > - The second one is the form of data, i.e., what is the "top-k" that the learner observes. As far as I can tell, the authors always assume k=1, i.e., the learner observes single-choice data. In other words, the top-k ranking model gives a tool to learn from single-choice data. Under this assumption, any discrete choice data should be suitable for testing. This is particularly true under the SUSHI data: the authors have already used a stylized way to convert the partial ranking data to single-choice data and "pretend" that they only see the single-choice data.
> > >
> > > - The third is the "top-k" in the "raw data" (before converting to single-choice data). This only applies because the authors use the SUSHI data, where the raw data has k = 10 (where n = 10 or n = 100). Again, those are converted to single-choice data before learning. This should not be confused with the model parameter described above. They are conceptually decoupled.
> > >
> > > (In fact, the authors' own experiment confirms decoupling: k = 10 in the raw SUSHI data, but the k inferred from the data is 14. This may initially appear contradictory, but it is not — it simply reflects the fact that the top-k parameter in the model is not constrained by the ranking length in the raw data, especially after the authors have transformed them into single-choices.)
> > >
> > > I hope the distinctions outlined above clarify the motivation behind my original Q3.

---

> > > > ### Author Response · Authors · 2025-08-05
> > > > **Re Re Rebuttal**
> > > >
> > > > **1**
> > > >
> > > > In the new model, when $k=n$ the runtime of DypCHIP is $\Theta(n^3 |{\mathcal A}|^2)$. This is because in equation 5, instead of summing over all $S\subseteq [k]$ we will have a sum over all subsets of $[n]$ with size at least $n$; thus the sum is over only one profile, namely $[n]$.So in fact, we don’t have $k2^k$ anymore rather just $1$ profile. As explained above, for large $k = n- \\ell $ we have $\ell n^\\ell$, and not $k 2^k$. In general for such $k$, in equation 5 we have to take a sum over all subsets of $[k]$ with size at least $k-\ell$ as there are at most $\ell$ elements that are not in top-$k$ and can be added to the profile to make a feasible top-$k$ ranking. Does this answer your question?
> > > >
> > > > **2**
> > > >
> > > > We thank the reviewer for articulating this distinction. You are correct, $k$ is a parameter of the top-k model, and for the observed data we use single choices. Since we did not find any dataset with pairs of assortments and single choices, we synthetically generated it from the sushi data set with $n=100$. We chose this dataset since it has the form of a top-k and we could use our own definition of choice for top-$k$. However, we agree with the reviewer that this is not necessary, and we could also use the sushi dataset with full rankings as raw data to curate the single choice data.
> > > >
> > > > We will make necessary changes in the paper to clarify this, and expand our experiments using full rankings in the extended version of this paper. Thank you.

---

> > > > > ### Comment · Reviewer_wgQk · 2025-08-06
> > > > >
> > > > > Thanks for the responses. Comment 1 addressed my concern about choice probability calculation. So it seems that the "hard" case is only when k is in the middle, rather than k is small or k is large.

---

> > > > > > ### Author Response · Authors · 2025-08-06
> > > > > >
> > > > > > It is true, when $k=n/2$  the preprocessing stage of PRIM and DyPCHIP have highest runtime complexity.
> > > > > >
> > > > > >
> > > > > > Thank you for the engaging discussion. We will incorporate your feedback in preparing the camera-ready version. Thanks.

---

> > > > > > > ### Comment · Reviewer_wgQk · 2025-08-07
> > > > > > >
> > > > > > > Thank you for the further responses. Overall, I found the author's rebuttal to be clear and effective, and I plan to raise my score.
> > > > > > >
> > > > > > > One final comment: in light of the discussion above (e.g., the distinction between different meanings of “top-k”), I found the title somewhat confusing, at least on first read. While the model proposes a way to describe and analyze a probability distribution over top-k lists, the use case in the paper is learning from single-choice data (i.e., "top-1"). Towards that direction, I believe some clarification in the paper would be helpful.

---

> > > > > > > > ### Author Response · Authors · 2025-08-07
> > > > > > > >
> > > > > > > > Thank you for the discussion. We appreciate your feedback and will clarify our text for better understanding. Thanks!

---

### Official Review · Reviewer_CcCu · 2025-07-03

**Clarity:** 4
**Significance:** 4
**Originality:** 4
**Rating:** 5
**Confidence:** 1

**Summary:**

This paper proposes a novel sampling strategy, an efficient algorithm, and a parameter learning method for generalized top-k Mallows models. Specifically, traditional Mallows models only fit the full permutation circumstances, while practical scenarios require Mallows models for top-k assortments, which are called generalized Mallows models. The proposed method extends the generalized Mallows models from three perspectives: RIM for generalized Mallows models, choice probabilities calculation for top-k list, and the choice centers construction. The authors provide the theoretical guarantee and empirical validation to support the effectiveness.

**Questions:**

1. How to use the proposed method for practical problems? In other words, can the authors provide a usage example/algorithm (or maybe it is one of the presented algorithms in the Appendix)?

2. How do centers of choices help to enhance the framework?

**Ethical Concerns:**

["NO or VERY MINOR ethics concerns only"]

**Final Justification:**

Due to the limited relevance to my area, other reviewers may ignore my ratings.

**Limitations:**

Yes.

**Quality:**

4

**Strengths And Weaknesses:**

I am totally unfamiliar with the topic and could only provide some reviews on formatting and illustration. My understanding of technical details might be wrong, and the authors are welcome to figure out mistakes within my review.

# Strength

1. Clarity. The authors clearly state their motivation, method, and experimental results. The background and notations for the studied problem, as well as the practical usage scenarios, are sufficiently illustrated to understand the problem and setting of the work.

2. The authors provide theoretical guarantees for the proposed method.

3. Extensive experimental results are provided to support effectiveness.



# Weakness

1. The text in several figures is too small to read.

2. The grammar in some places might be problematic; please check. For example, around Line 23, the authors wrote, "Based on Plackett-Luce (PL) model the Multinomial Logit (MNL) model has been suggested", where there should be a comma between "model" and "the".

3. A holistic comparison with current methods may help to signify the importance of the proposed method. The authors illustrated their contributions by iterating the components, where there is a lack of an overview of the systematic advancement of current methods. The authors may illustrate how they achieve more usable/suitable generalized Mallow models by proposing three novel components.

---

> ### Author Rebuttal · Authors · 2025-07-30
>
> Thank you for the supportive review. We appreciate the careful read and make sure to fix the typos for the final version. Please see below for responses to your questions:
>
> **Q1**
>
> The proposed methods, centered around the Generalized Top-k Mallows Model (TopKGMM), offer a robust framework for addressing practical problems involving user preferences, especially when those preferences are expressed as top-k lists rather than full rankings. For practical application, the process primarily involves two key algorithms: BUCCHOI and DYPCHIP. BUCCHOI is used first to learn the underlying central ranking ($\tau^\*$) and other model parameters from observed customer choices, which are typically recorded as (assortment, chosen_item) pairs. It achieves this by adaptively presenting specific assortments and inferring preferences based on selections. Once the model parameters are established, DYPCHIP efficiently calculates the choice probability for any given item within an assortment. This predictive capability is vital for businesses, enabling them to optimize product assortments, refine recommendation systems, and enhance advertising strategies.
>
> **Q2**
>
> We are not sure if we understand this question clearly. Please let us know if the following answers your question.
>
> The main ranking acts as the fundamental "true" preference order for a user or a group of users. Rather than modeling preferences as a set of disconnected probabilities, the center provides a single, intuitive ranking that summarizes the core preference structure. This makes the model more interpretable; one can look at the central ranking to understand the typical preference of a customer population.

---

### Official Review · Reviewer_W9Hi · 2025-07-03

**Clarity:** 2
**Significance:** 3
**Originality:** 3
**Rating:** 4
**Confidence:** 2

**Summary:**

The authors propose a generalized Mallows model for top-k lists and develop efficient algorithms
related to generating samples from it, learning its parameters and finding choice probabilities.

In particular, they propose a sampling scheme with running time exponential in k (the length of the top-k list) and polynomial in n, improving upon the previously best sampling scheme, and adapted to the generalized top k Mallows model. They give a dynamic programming algorithm to compute the choice probabilities
And finally provide a method to learn the center of the top-k generalized Mallows model.

Experimentally, they check the runtime of their algorithms and also show that that the test error on the Sushi data set is lower as compared to the Multinomial Logit Model.

**Questions:**

Why is Generalized Mallows sensible? Why would you only have weights attributed to the higher element?
Is this well motivated or does it simply make the math easier?

What is the sample complexity of doing the same things as you do in the paper for the Plackett-Luce model? In particular, the Plackett Luce model seems mode amenable to top-k rankings, since its sampling algorithm exactly sampled in order of ranking.What is the main different between Plackett Luce and Mallows, especially if in Plackett Luce once uses exponential weights?


Minor Questions:

Why do you say that two items in tau bar are incomparable as opposed to assuming they are a tie (mathematically)?
Line 29: "High predictive Power of the Mallows model...": What do you mean by this?



Line 33: Can you reference concrete examples where platforms show user specific top k relevant items and the k
Is the same across all users?

What is \tau^* in Equation 1? Do you mean \tau'?

**Ethical Concerns:**

["NO or VERY MINOR ethics concerns only"]

**Final Justification:**

The authors resolved several concerns I raised and promised to improve the clarity of the paper.

**Limitations:**

The algorithms are exponential in the parameter k, thus not practicable for larger k. It is not clear why the generalized version for top-k Mallows is needed.

**Quality:**

2

**Strengths And Weaknesses:**

I don't feel particularly convinced by several parts of the paper and I feel that I struggle to understand its relevance -  please see my questions for several things I am unclear about. The authors could much improve the motivation in the paper by offering concrete examples in which their particular model is more useful/ important/ relevant than other ranked choice models.

The experiments are down for the top-k mallows model not its generalized version and thus do not support the version introduced by the authors in this paper . This begs the question where one would use the weighted version.
Also the TopKGMM mixture models are computed by using the Kendall-Tau distance K^p, and then used to compare TOPKGMM with MNL, and I worry that this inherently is advantageous towards TOPKGMM. At the very least the authors should discuss this choice. Figure 1 and the meaning of the test error is further not clear to me.

I am not convinced that this paper solves an open question regarding a new RIM for top-k Mallows. My impression would be that RIM is a very nice and simple method, and something is RIM-like in so far as it is still simple, not because it uses insertions in some way.

Line 164:
I would be useful to explain in words to the reader what I, P and Q mean.
For P_3(tau): are not 4,7,8 all greater than 3 and all of 3,4,7,8 satisfy that they ar in \bar(\tau)? Thus, shouldn't
P_3(\tau)=3? Similarly P_4(\tau) seems wrong?
Line 45: "efficient" - how is exponential in k an efficient algorithm?
Line 159: Should the index in the summation include \tau^*? What is ">"? Is it the same as the curly version?

Minor comments:

Line 158: It would be helpful for the reader if you introduced the general inversion vectors before you introduce the special case, to understand what they are.
Line 115: Chioces -> Choices
Line 124: full stop missing
Line 163: "For example,.." This is not a full sentence
Line 167L "," should be on the previous line
Line 210: later -> latter

Line "219": One cannot evaluate the computational complexity of an algorithm through experiments..

---

> ### Author Rebuttal · Authors · 2025-07-30
>
> Thank you for your time and careful reading of our paper. We try our best to answer your questions and looking forward to constructive conversation. We will fix the typos and apply the suggestions for clarifying the paper.
>
> **[Q1-a] Why is Generalized Mallows sensible?**
>
> Generalized Mallows (for permuations) is introduced by Fligner & Verducci and it is a sensible extension because it more realistically captures user preferences by acknowledging that not all items are equally important. We develop this same extension to Chierichetti et al work on top-k lists .
>
> **[Q1-b] Why would you only have weights attributed to the higher element? Is this well motivated or does it simply make the math easier?**
>
> The weights are associated with the “rank” of items. The higher elements in the top-k have rank $1,2,\dots k$ and the lower elements all have rank $k+1$. This is consistent with the applications in which platforms have top-k rankings and the users are not exposed to the lower items.
>
>
> **[Q2] What is the sample complexity of doing the same things as you do in the paper for the Plackett-Luce model? In particular, the Plackett Luce model seems more amenable to top-k rankings, since its sampling algorithm exactly sampled in order of ranking. What is the main difference between Plackett Luce and Mallows, especially if in Plackett Luce once uses exponential weights?**
>
> Models based on Plackett-Luce, such as the Multinomial Logit (MNL) model are inherently different from Mallows models. The mallows model introduces a distribution based on the distance of a permutation (or a  top-k list) from the center distribution (or center top-k list); this distance is defined in terms of the number of transpositions (swaps) to go from one permutation to another one. Plackett-Luce Model does not use a central ranking or a distance metric. Instead, it assigns an intrinsic positive score or utility to each item. The probability of a ranking is determined by a sequential choice process: the probability of picking an item first is its score divided by the sum of scores of all available items. The process repeats for the remaining items. Using exponential weights (e.g., as in the MNL formulation) does not change this underlying score-based mechanism.
>
> Another main difference of  PL/MNL model and Mallows models is in satisfying Independence of Irrelevant Alternatives (IIA) axiom which means the relative probability of choosing item $i$ over item $j$ is unaffected by the presence of a third item $\ell$. The Mallows model violates the IIA axiom while MNL satisfies this axiom. In Mallows model, the probability of preferring $i$ over $j$ depends on the entire central ranking $\tau$. This allows the Mallows model to capture more complex and realistic substitution effects among items, which is a key reason for its high predictive power.
>
>
> We now respond to minor questions (MQ):
>
> **[MQ1] Why do you say that two items in a tau bar are incomparable as opposed to assuming they are a tie (mathematically)?**
>
> Could you please clarify this question? When saying i and j are incomparable we mean that the user has no preference of one over the other. As in previous work, e.g., Chierichetti et al, this is the terminology that is used. This shows a scenario in which users have no preference over less relevant items and they are ranked lowest for them.
>
>
> **[MQ2] Line 29: "High predictive Power of the Mallows model...": What do you mean by this?**
> We mean that Mallows model when compared to MNL offers higher prediction accuracy (i.e., achieves lower test error) on several publicly available datasets. This is demonstrated by the work of Desir et al for the classic Mallows model and we show it here for top-k MM on Sushi data set.
>
>
> **[MQ3] Can you reference concrete examples where platforms show user specific top k relevant items and the k Is the same across all users?**
> In many real-world applications, platforms have a fixed and limited space to display items. This is particularly true in online settings where, for example, a dedicated area on a website or an advertising box, e.g., on Google or Facebook, can only fit a predetermined number of choices. It also happens in the physical world, for example display areas of stores that display the latest arrivals however it has a fixed setup that would fit a predetermined number of items, e.g., stands in the store with a fixed number of shelves. Because of these spatial limitations, users are typically shown only the top-k most relevant items rather than an exhaustive list. This approach aligns well with user behavior, as people often express clear preferences for a limited number of favorite items and are indifferent toward the rest. Consequently, the preference data that can be collected from these platforms naturally takes the form of top-k lists instead of full rankings.
>
> **[MQ4] What is $\tau^\*$ in Equation 1? Do you mean $\tau'$?**
>
> Thanks for flagging this. There is a typo in the equation. We use $\\tau^\*$ to refer to the central ranking, and we changed Eq (1) to make it a general definition and not just with respect to $\tau^\*$. However, we seem to have forgotten to completely update it. $\\tau^\*$ should be updated to $\\tau’$ in the formula. We have updated our manuscript to reflect this.
>
> We now responds to points made in Weaknesses:
>
> **[W0] What is the main different between Plackett Luce and Mallows, especially if in Plackett Luce once uses exponential weights?**
>
> Models based on Plackett-Luce, such as the Multinomial Logit (MNL) model are inherently different from Mallows models. The mallows model introduces a distribution based on the distance of a permutation (or a  top-k list) from the center distribution (or center top-k list); this distance is defined in terms of the number of transpositions (swaps) to go from one permutation to another one. Plackett-Luce Model does not use a central ranking or a distance metric. Instead, it assigns an intrinsic positive score or utility to each item. The probability of a ranking is determined by a sequential choice process: the probability of picking an item first is its score divided by the sum of scores of all available items. The process repeats for the remaining items. Using exponential weights (e.g., as in the MNL formulation) does not change this underlying score-based mechanism.
>
>
> Another main difference of  PL/MNL model and mallows models is in satisfying Independence of Irrelevant Alternatives (IIA) axiom which means the relative probability of choosing item $i$ over item $j$ is unaffected by the presence of a third item $\ell$. The Mallows model violates the IIA axiom while MNL satisfies this axiom. In Mallows model, the probability of preferring $i$ over $j$ depends on the entire central ranking $\tau$. This allows the Mallows model to capture more complex and realistic substitution effects among items, which is a key reason for its high predictive power.
>
>
> **[W1] The experiments are down for the top-k mallows model not its generalized version and thus do not support the version introduced by the authors in this paper. This begs the question where one would use the weighted version**
>
> We extended the topKMM model to topKGMM model since our sampling and dynamic programming algorithms are applicable in the general setting.
> Our learning algorithm is also applicable to arbitrary weights as long as the condition of $\beta>\log 3/w_{\min}$ is satisfied (see theorem 5.1). The learning algorithm learns the center as well as the parameter $k$ and does not learn the weight vector $\vec{w}$; learning weights is a difficult problem and is left as a future direction.
> Various weight vectors (combination of 3s 2s and 1s) are chosen in synthetic experiments. For the experiments on sushi data we only used the topkMM model since we have no algorithm for learning the weights.
>
> **[W2] Figure 1 and the meaning of the test error is further not clear to me.**
>
> We have run the algorithms 10 times and the figure shows the mean and std of the test error.
>
> **[W3] I am not convinced that this paper solves an open question regarding a new RIM for top-k Mallows. My impression would be that RIM is a very nice and simple method, and something is RIM-like in so far as it is still simple, not because it uses insertions in some way**
>
> A key property of RIM is that it inserts the elements iteratively in the permutation (top-k list) by explicitly calculating the probabilities of insertion. This property is fundamental to the design of the dynamic programming algorithm for calculating choice probabilities.
> Our algorithm PRIM has a similar property: after sampling a profile inserts the element iteratively in the top-k by explicitly calculating the probabilities of insertion. This property is used in the design of DyPHIP.
>
> **[W4] Clarification request**
> Thanks for flagging the areas that need clarification. We will make sure to address these in the final version.

---

> > ### Author Response · Authors · 2025-08-05
> >
> > Dear Reviewer W9Hi,
> >
> > We are wondering if we have addressed your concerns in the rebuttal. We would be happy to engage in an interactive discussion to clarify any remaining questions.

---

> > ### Comment · Reviewer_W9Hi · 2025-08-07
> >
> > I thank the authors for addressing most of my concerns! I will increase my score.
> > 1. Could you clarify my question from my review:
> > "For P_3(tau): are not 4,7,8 all greater than 3 and all of 3,4,7,8 satisfy that they are in \bar(\tau)? Thus, shouldn't P_3(\tau)=3? Similarly P_4(\tau) seems wrong?"
> > 2. Could you also elaborate in which practical applications a top-k Mallows model vs a top-k PLM should be used?

---

> ### Author Response · Authors · 2025-08-07
> **Re remaining questions from rebuttal**
>
> Thank you for your questions, let us elaborate these two points a bit further.
>
> **1**
>
> We have $\tau^\* = 1 \\succ 2 \\succ 3 \\succ 4 \\succ \\{5,6,7,8\\}$ meaning that $1$ is the most preferred item, then $2$, then $3$, and then $4$, and $\\{5,6,7,8\\}$ are not ranked. When we count the number of comparable and incomparable pairs, we consider $i,j\in \tau\cup \tau^\*$. In fact, in the section under discussion, our goal is to show how we can rewrite the equation (1), where only elements in $ \tau\cup \tau^\*$ matter. Since elements $7$ and $8$ are not ranked by either of $\tau$ and $\tau^\*$, they don’t contribute to $P_3$. Therefore, in the inversion table of 3, we are **not** comparing it with 7 and 8. The only pair that gets counted in $P_3$ is $4$ and we have  $3\\succ_{\tau^\*} 4$ and $3\parallel_{\tau} 4$. We are happy to clarify this better in our camera ready version.
>
> **2**
>
> The Mallows model, analogous to the Gaussian distribution, is defined on the space of rankings, whereas the Gaussian distribution is defined on $\\mathbb{R}$. Both models posit a central value—a mean in the Gaussian context and a central ranking in the Mallows model—from which observed data are assumed to be distorted by noise. For instance, in the Gaussian case, a true scalar value of 3 might yield observations like $3.01$, $3.05$, or $2.95$, each with a probability derived from the Gaussian distribution. Similarly, in the Mallows model, a true top-$k$ ranking of $1 \\succ 2 \\succ 3 \\succ 4$ could result in observed rankings such as $2 \\succ 1 \\succ 3 \\succ 4$, $1 \\succ 2 \\succ 4 \\succ 3$, or $5 \\succ 1 \\succ 2 \\succ 3$, with probabilities determined by the Mallows distribution. In both scenarios, the probability decreases as the distance from the center increases.
>
> In contrast, the Plackett–Luce model lacks a **true central value**; instead, each item is associated with a utility value. Consequently, in the Mallows model, the probability of an item appearing at a specific location in a ranking is **dependent** on other elements, while in the Plackett-Luce model, independence holds.
>
> The Mallows model is applicable in any scenario where a true ranking can be assumed and observations are subject to noise. For example, a driver may have a preference over available routes from work to home (e.g., $A \\succ B \\succ C \\succ D$), but varying weather conditions, departure times, or interim errands might lead to observations of alternative rankings. Similarly, a customer's true ice cream preference ($A \\succ B \\succ C \\succ D$) might be obscured by situational preferences, resulting in observed rankings such as $B \\succ A \\succ C \\succ D$ or $A \\succ C \\succ B \\succ D$. This illustrates the concept of observed data being distorted from true values.
>
> Our research, along with that of Desir et al., demonstrates that the Mallows model exhibits superior predictive power for the sushi dataset, a widely used benchmark for ranking data.
>
> Please let us know if this sufficiently addresses your concerns, and if we can provide any further clarification.

---

> > ### Author Response · Authors · 2025-08-08
> >
> > Dear Reviewer W9Hi,
> >
> > As the author reviewer discussion period is ending, could you please let us know if we have addressed your concerns.
> > Thanks.

---

> > ### Comment · Reviewer_W9Hi · 2025-08-08
> >
> > Thank you for clarifying my misunderstanding in the example. I realize I was confused about the computations of the inversion vectors since on first reading the definitions are somewhat involved. Thanks also for clarifying some of my question regarding Mallows vs PL. I suppose I am still looking for a more comprehensive (beyond a single data set like Sushi) comparison of the two model on real-world data.
> >
> > Since most of my concerns were clarified, I will raise my score!

---

### Note · Authors · 2025-08-12

We are glad that reviewers were happy with our answers and appreciate their engagement, which has been invaluable in identifying sections that require further clarification. Namely,

**Top-k**

Reviewer wgQk noted that the term "top-k" is used in various contexts and our usage has been ambiguous. We will clarify that "top-k" refers to a partial order where the top k elements are ranked linearly and TopkGMM refers to mallows model on top-k lists. While "top-k" sometimes refers to the top *choices* by a customer in the literature, we use "choice" to denote a *top-1* preferred item in our paper.

**learning algorithm BuCCHOI**

Reviewers raised the following questions

> Is BuCCHOI a passive or active learning algorithm?

BuCCHOI is an active learning algorithm that strategically selects assortments and collects choice data. FINDTOP can be used on historical data for passive learning.

> Which parameters are learned by BuCCHOI?

BuCCHOI learns the center and the parameter k.

> How crucial is the usage of pairwise comparisons in design of BuCCHOI?

While pseudocode used size-two assortments for simplicity, the idea extends to arbitrary size $\ell$ assortments to learn the center and k. Pseudocode and analysis for arbitrary-sized assortments will be included in the paper.


**prior work**

Reviewer wgQk inquired about comparisons to prior work on pairwise comparisons. We will cite Lu and Boutilier and Vitelli et al., which use Maximum Likelihood Estimation. Our work, which presents finite sample complexity and error bounds, aligns more with Tang ICML 2019 (classic Mallows model) and Feng and Yuxuan NeurIPS 2022 (Mallows-like distribution with a non-inversion-based distance measure).

Reviewer jMHU asked about comparison with Fotakis et al which focuses on aggregating incomplete and noisy rankings to find a consensus, a distinct task from learning model parameters using single choices from assortments. We believe extending Fotakis et al. to learn a center based on single choices is a promising area for future research.

**using other data sets**

We used the top-10 sushi dataset due to its prominence in customer preferences. While reviewers suggested using full ranks or other datasets, we will expand our experiments to include these in the paper's extended version.

**inversion tables**

Per W9Hi's note, we will present inversion tables more clearly.

**presentation**

We will increase the font sizes in the tables and figures and correct the flagged typos.

---

### Decision · Program_Chairs · 2025-09-17

**Decision:**

Accept (spotlight)

**Comment:**

This study examines top-$k$ Mallows models for preference modeling and presents algorithmic contributions related to generating samples, determining choice probabilities, and learning model parameters. It includes a thorough analysis of the runtime complexity of these algorithms, supported by experimental results. Overall, the paper has received five positive reviews, with an average score reflecting a strong likelihood of acceptance. Therefore, I recommend it for acceptance.

Additionally, the authors have effectively addressed concerns about the clarity and significance of the study during the rebuttal phase. I suggest incorporating these improvements into the revised version of the paper.